# Distinct iron cycling in a Southern Ocean eddy

Michael J. Ellwood [ID] [1]*, Robert F. Strzepek [ID] [2], Peter G. Strutton[2,3], Thomas W. Trull[4], Marion Fourquez[2] & Philip W. Boyd [ID] [2]

Mesoscale eddies are ubiquitous in the iron-limited Southern Ocean, controlling ocean-atmosphere exchange processes, however their influence on phytoplankton productivity remains unknown. Here we probed the biogeochemical cycling of iron (Fe) in a cold-core eddy. In-eddy surface dissolved Fe (dFe) concentrations and phytoplankton productivity were exceedingly low relative to external waters. In-eddy phytoplankton Fe-to-carbon uptake ratios were elevated 2–6 fold, indicating upregulated intracellular Fe acquisition resulting in a dFe residence time of ~1 day. Heavy dFe isotope values were measured for in-eddy surface waters highlighting extensive trafficking of dFe by cells. Below the euphotic zone, dFe isotope values were lighter and coincident with peaks in recycled nutrients and cell abundance, indicating enhanced microbially-mediated Fe recycling. Our measurements show that the isolated nature of Southern Ocean eddies can produce distinctly different Fe biogeochemistry compared to surrounding waters with cells upregulating iron uptake and using recycling processes to sustain themselves.

[1] Research School of Earth Sciences, Australian National University, Canberra, Australia. [2] Institute for Marine and Antarctic Studies, University of Tasmania, Hobart, Australia. [3] Australian Research Council Centre of Excellence for Climate Extremes, University of Tasmania, Hobart, Australia. [4] CSIRO Oceans and Atmosphere, Hobart, Australia. *email: Michael.ellwood@anu.edu.au

Mesoscale eddies are ubiquitous in the ocean[1,2] and play a crucial role in the transfer of heat, carbon and nutrients between the deeper ocean, surface waters and the atmosphere[3–6]. Cold-core eddies in the Southern Ocean are defined by strong clockwise rotation, cooler temperatures and negative sea-surface height anomalies[7,8]. These eddies can have closed circulation thus trapping[9] the biogeochemical properties of these features such that nutrient, chlorophyll and particle concentrations can be distinct relative to those in the surrounding waters[2,9–11]. They can transport these biogeochemical properties vast distances[12] thus they are important from an oceanographic point of view, especially if they cross water mass boundaries such as the polar front or the subantarctic front[7,11,13].

The concentration of dissolved Fe (dFe) in remote Southern Ocean surface waters, away from continental and island input sources, is typically sub-nanomolar (60–200 pmol kg$^{-1}$)[14,15]. The lower limit for this dFe range is thought to be controlled by organic complexation and atmospheric supply. In the Southern Ocean, atmospheric inputs are very low and the supply of Fe usually is provided via upwelling and upward mixing of deeper iron-enriched waters[14,16]. However, eddies can become 'structurally closed' post-development, so their ability to entrain and detrain dFe, nutrients, phytoplankton and zooplankton can be restricted[1,17], thus making them a static mesocosm-like environment. Structural breakdown of the eddy and the export of sinking organic matter are the principal biogeochemical loss terms. Our study focused on one such eddy to understand the biogeochemical cycling of Fe in this isolated structure. We contextualise our in-eddy Fe speciation (dissolved and particulate) and isotope measurements with primary productivity and biological observations and contrast them to external sites within the subantarctic zone.

## Results and Discussion

**Eddy development and characterisation.** Between 30 March and 4 April 2016, we sampled a cyclonic cold core eddy that had spawned from the northern jet of the subantarctic front in mid-February 2016[3,4]. Satellite-derived sea surface temperature imagery, vertical profiles of temperature, salinity and nutrients, and undulating sensor measurements revealed that the eddy was biogeochemically distinct from the waters surrounding it at two reference stations (SAZ and SOTS; Fig. 1, Supplementary Figs. 1, 2 and Supplementary Movie 1). This eddy was typical with respect to its size (diameter) and life-span with that of other eddies in the region[3], however it was more intense in terms of rotational speed and amplitude. Sea surface temperatures in the eddy were about 2 °C lower than surrounding waters, again more intense than basin-averaged anomalies of about 0.5 °C[1]. Chlorophyll concentrations were approximately 1.5 times lower within the eddy, consistent with the analysis of trapping eddies[9,11]. Large differences in fluorescence and transmissivity with depth inside the eddy highlight low biological production at the time of sampling (Supplementary Fig. 1), which we confirmed with in situ biological measurements. In-eddy pico-plankton and nano-plankton cell numbers and primary productivity rates were reduced compared to external stations (Fig. 2, Supplementary Figs. 3, 4 and Supplementary Table 1). In-eddy primary productivity declined from 0.147 mmol C m$^{-3}$ d$^{-1}$ at 80% incident irradiance (15 m) to 0.018 mmol C m$^{-3}$ d$^{-1}$ at 1% incident irradiance (100 m). Outside of the eddy, primary productivity was higher, with values of 0.31 and 0.38 mmol C m$^{-3}$ d$^{-1}$ at 80% incident irradiance (15 m) at the SAZ and SOTS sites, respectively. This is consistent with productivity (~0.3 mmol C m$^{-3}$ d$^{-1}$) previously determined for the region[18], and about three times higher than in-eddy rates. These observations are all consistent with the eddy originating south of the northern extension of the Subantarctic Front where temperatures and chlorophyll concentrations were lower (Fig. 1 and Supplementary Movie 1)[4].

In the Southern Ocean, Fe limits phytoplankton production seasonally[16,18]. Our in-eddy dFe concentrations are amongst the lowest ever measured for the Southern Ocean, with values between 18 and 33 pmol kg$^{-1}$ for the upper 200 m (Fig. 3, Supplementary Fig. 2; Supplementary Table 2). Because the eddy developed in the northern jet of the Subantarctic Front, the Fe status of the eddy likely reflects the initial biogeochemistry of waters further south but modified on its northward transit. Indeed, nitrate and phosphate concentrations at 53°52'S; 147°16'E for waters near where the eddy formed were 2.3 μmol L$^{-1}$ and 0.1 μmol L$^{-1}$ higher, respectively than in-eddy levels suggesting biogeochemical modification of surface water properties as the eddy evolved. Interestingly, silicate levels were similar between

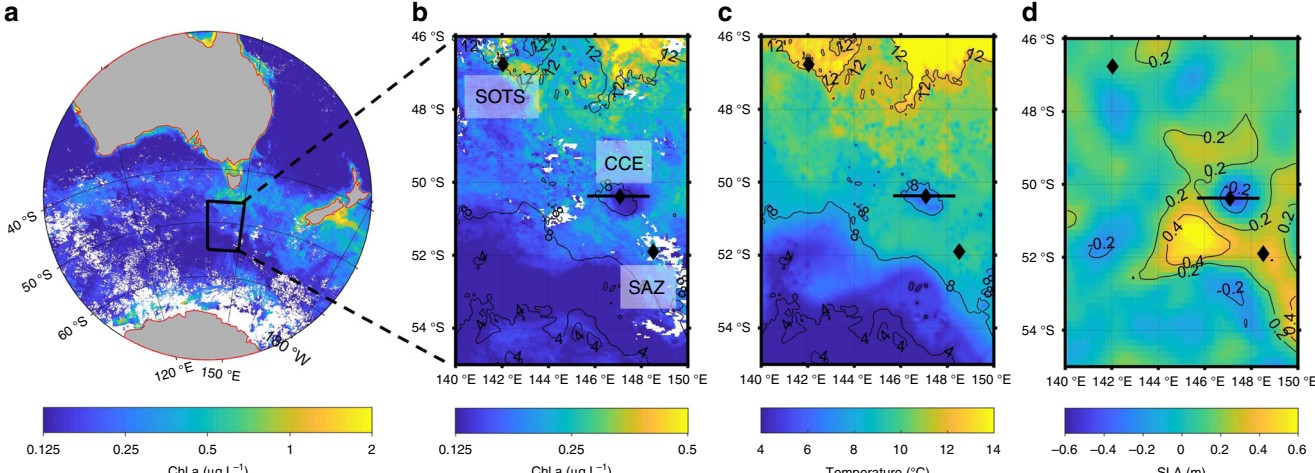

**Fig. 1 Chlorophyll and sea surface temperature in the study area. a** and **b**, Maps of chlorophyll *a* concentration **c**. Sea Surface Temperature (SST) and **d**. Sea Level Anomaly (SLA) for Southern Ocean waters south of Australia. The diamonds represent the Cold Core eddy station (CCE), the Subantarctic zone station (SAZ) and the Southern Ocean Time Series station (SOTS) and the solid black line represents the Triaxus tow (Supplementary Fig. 1). The chlorophyll *a* satellite data represents a monthly average for March 2016. The SST data are for 3 April 2016 and SLA data are for 25 March 2016. In **b** and **c** the contours lines are SST. Data extracted from https://coastwatch.pfeg.noaa.gov/erddap/griddap/.

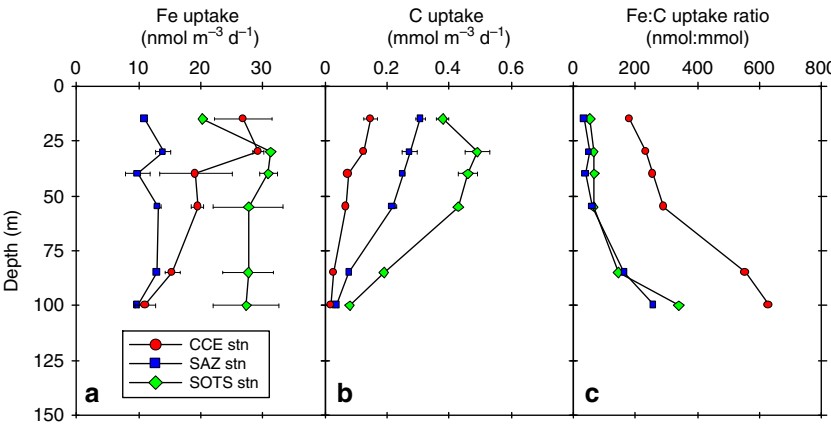

**Fig. 2 Depth profiles of Fe and carbon uptake.** Profiles of **a** Fe and **b** carbon uptake versus depth. **c** Profiles of the intracellular Fe:C uptake ratio for the CCE, SAZ, and SOTS stations. Error bars represent 1 s.d. for replicate measurements.

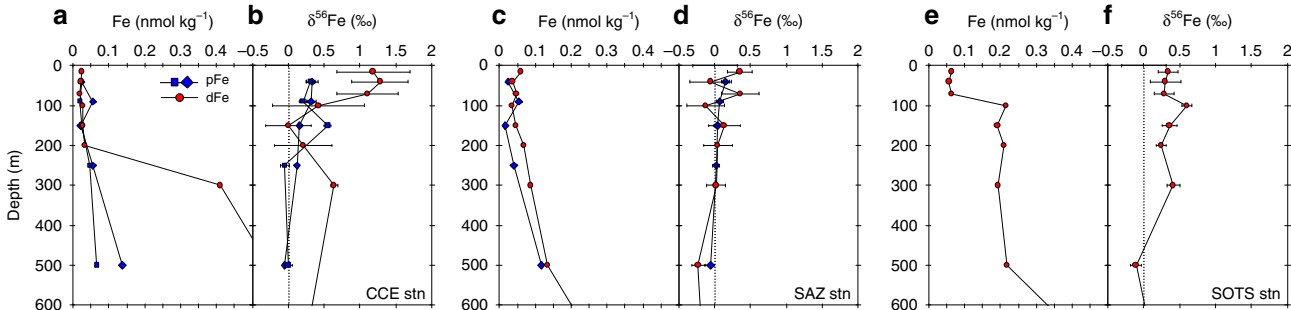

**Fig. 3 Depth profiles of Fe concentration and isotope composition.** Upper ocean (0–600 m) depth profiles of dFe and pFe concentration (**a–c**), and the isotope composition (**d–f**) for samples collected at the **a**, **d** CCE, **b**, **e** SAZ and **c**, **f** SOTS stations. See the Methods section for details on the issues and challenges involved with measuring δ56Fe on samples with very low dFe concentrations. Error bars for isotope measurements represent 2 s.e.m.

the two sites suggesting that changes in nitrate and phosphate drawdown are likely facilitated by non-siliceous phytoplankton. Previous nearby summertime measurements for dFe from near where the eddy spawned, at and south of the Subantarctic Front, ranged between 70 and 210 pmol kg$^{-1}$, thus these low in-eddy dFe values appear to result from in-eddy processes (Table 1). Outside the eddy, dFe concentrations in the upper 200 m were 35–68 and 57–210 pmol kg$^{-1}$ for the SAZ and SOTS stations, respectively (Fig. 1), consistent with measurements for this region and the Southern Ocean generally (Table 1)[15]. Like dFe, in-eddy particulate Fe (pFe) concentrations were extremely low with values between 13 and 29 pmol kg$^{-1}$ for the upper 200 m (Fig. 3; Supplementary Table 3). These low pFe concentrations reflect the low input of lithogenic material (dust) into the Southern Ocean (see Supplementary Information)[19]. This observation is supported by the fact that between 51 and 69% of the pFe pool in the upper 250 m of the eddy is considered biogenic (Supplementary Fig. 5).

**Iron and carbon uptake and dFe residence time.** Our results raise the question: How can dFe be consumed to such low concentrations by the in-eddy phytoplankton community, compared to external waters? We explored this question by measuring the rate of Fe uptake by cells inside and outside of the eddy. In-eddy Fe uptake rates were higher at 15 m (80% incident irradiance) compared to rates outside the eddy at the two reference sites (Fig. 2 and Supplementary Fig. 2, Supplementary Table 4). With depth, in-eddy Fe uptake rates decreased, but they were always higher than rates measured for the SAZ site (Fig. 2). In-eddy carbon-normalised Fe uptake rates (expressed as a Fe:C ratio) for

the resident phytoplankton community were approximately 4-fold higher compared to the two references sites and historical measurements for the region (Table 1). This is a surprising result considering the low in-eddy biomass at the time of occupation (Supplementary Figs. 1, 3). A higher Fe:C uptake ratio for in-eddy phytoplankton is consistent with upregulation of cellular Fe acquisition machinery, to help acquire dFe. Indeed, Fe-limited phytoplankton have been shown to transport Fe into the cell at a faster rate than Fe-replete cells[20]. The structurally isolated nature of the eddy with respect to deep and surrounding waters (Supplementary Fig. 1) also reduced dFe replenishment from below and laterally. The diffusional supply of dFe into the euphotic zone (0–85 m) is estimated to be 0.25 nmol m$^{-2}$ d$^{-1}$ based on the dFe gradient between 200 and 70 m and an eddy diffusion coefficient of $K = 3 \times 10^{-5}$ m$^2$ s$^{-1}$[16,21]. This diffusional dFe supply rate is approximately $7 \times 10^3$ times lower than the Fe utilisation rate of 1.88 μmol m$^{-2}$ d$^{-1}$ across the euphotic zone, thus highlighting the extreme nature of this cold core eddy with respect to Fe supply. It is possible that $K$ values in this region might be as high as $10^{-4}$ m$^2$ s$^{-1}$, but values higher than $10^{-3}$ m$^2$ s$^{-1}$ are never observed[22] and for the diffusive supply and biological demand to match would require $K = 0.2$ m$^2$ s$^{-1}$. Using the utilisation rate and a dFe inventory of 1.87 μmol Fe m$^{-2}$ for the euphotic zone, we estimate a residence time of 1 day for dFe. This short residence time indicates that Fe is being heavily trafficked within the euphotic zone between the dissolved pool and the microbial community. The increased importance of Fe recycling favours smaller phytoplankton cells, which is reflected in the cell abundances, the size-fractionated iron uptake and the Fe:C ratio datasets (Supplementary Figs. 3, 4).

**Table 1 Intracellular iron:carbon (Fe:C) ratios, Fe uptake rates and dissolved Fe concentrations for upper Southern Ocean waters between longitudes 142°E and 172°E.**

| Station/Study latitude | Fe:C ($\mu$mol mol$^{-1}$) | Fe uptake rate (pmol L$^{-1}$ d$^{-1}$) | Dissolved Fe (pmol kg$^{-1}$) | Notes | Reference |
|---|---|---|---|---|---|
| Cold core eddy—49.7°S | 224 ± 38 | 25 ± 5 | 23 ± 1 | 15–40 m—80–20% irradiance | This study |
| SAZ—51.4°S | 40 ± 8 | 12 ± 2 | 48 ± 17 | 15–40 m—80–20% irradiance | This study |
| SOTS—46.7°S | 61 ± 7 | 28 ± 6 | 60 ± 4 | 15–40 m—80–20% irradiance | This study |
| SOIREE—61°S | 3–7.5# | 3.07# | 80 ± 30$ | Fe enrichment experiment | Bowie et al.[53] |
| FeCycle—46°S | 5.5 - 19 | 290–360 | 51 ± 11 | Subantarctic experiment | McKay et al.[54] |
| SOFex—56°S | 9–11 | not reported | 140 | Northern Patch | Twining et al.[55] |
| SAZ Project—47°S | 52 | 60 | 70 | SOTS station | Boyd et al.[18] |
| SAZ Project—54°S | 78 | 55 | 70 | Polar water station—equivalent to where the cyclonic eddy originated | Boyd et al.[18] |
| SAZ-Sense P1—46.3°S | 70 ± 44 | 110 ± 10 | 260 ± 40 | Equivalent to SOTS station | Bowie et al.[56] |
| SAZ-Sense P2—54°S | 60 ± 9 | 34 ± 5 | 210 ± 20 | Polar water station—equivalent to where the cyclonic eddy originated | Bowie et al.[56] |
| SAZ-Sense P3—45.5°S | 74 ± 47 | 77 ± 10 | 440 ± 70 | Site close to Subtropical convergence | Bowie et al.[56] |

$Background dissolved Fe concentration before Fe infusion.
#Fe:C and Fe uptake rates are for days 1–3 after Fe infusion.

**Iron acquisition and recycling processes**. The elevated in-eddy Fe:C uptake ratios also raise the question as to how phytoplankton are enhancing dFe uptake. Enhanced dFe uptake can occur through a combination of processes[14], including increased production of Fe transporters on the surface of cells, a reduction in cell size, the production of Fe binding ligands, and the use of Fe (III) reductase proteins to enhance Fe(II) production and hence the acquisition of Fe from organic complexes. We used the isotopic composition of dFe and pFe to probe Fe uptake within the eddy.

The isotope composition of dFe ($\delta^{56}$Fe$_{diss}$) showed distinct variability with depth and between stations (Fig. 3). In the euphotic zone (0–85 m) of the cold core eddy, $\delta^{56}$Fe$_{diss}$ values are isotopically heavy at +1.19‰ and distinct ($p < 0.005$, T-test) to that of pFe ($\delta^{56}$Fe$_{part}$) and the $\delta^{56}$Fe$_{diss}$ composition for waters below the euphotic zone (0.00 ‰ at 150 m). The $\delta^{56}$Fe$_{diss}$ composition for surface waters at this site are also isotopically distinct ($p < 0.005$) to $\delta^{56}$Fe$_{diss}$ values measured at the SAZ and SOTS stations (Fig. 3). At these stations, euphotic zone $\delta^{56}$Fe$_{diss}$ and $\delta^{56}$Fe$_{part}$ are isotopically similar to each other but significantly lower ($p < 0.005$) than in-eddy $\delta^{56}$Fe$_{diss}$ values (Fig. 3).

The heavy in-eddy $\delta^{56}$Fe$_{diss}$ values for the euphotic zone are consistent with biological fractionation during dFe acquisition by phytoplankton. Modelling of the $\delta^{56}$Fe$_{diss}$ dataset using a closed system model produced isotope fractionation factors ($\epsilon$) of −2.3‰ for samples collected in the euphotic zone (15–100 m; Fig. 4). Interestingly, the in-eddy $\delta^{56}$Fe$_{part}$ values for the euphotic zone are not consistent with an instantaneous or an integrated isotope fractionation process associated with a closed system model. Generally, the expectation is that as dFe is consumed, the pFe pool should become isotopically heavier for both the instantaneous and the accumulated product. While $\delta^{56}$Fe$_{part}$ does appear to be subtly heavier with decreasing dFe concentration, it is not consistent with closed-system dependency for the biological reduction of Fe(III) to Fe(II) by cells (Fig. 4). The physical cycling of $\delta^{56}$Fe$_{part}$ and $\delta^{56}$Fe$_{diss}$ offers a possible explanation for this discrepancy: $\delta^{56}$Fe$_{part}$ is distributed downward through the euphotic zone without general modification by sinking, whereas changes in the $\delta^{56}$Fe$_{diss}$ with depth (and dFe concentration) across the euphotic likely represent mixing across the euphotic zone.

Using a generalised 1-D model, we simulated these two different mechanisms over the seasonal cycle and found that the $\delta^{56}$Fe$_{diss}$ signal could be adequately modelled using isotope fractionation associated with just dFe uptake alone ($\epsilon = -1$‰) or a combination of isotope fractionation processes, namely dFe uptake ($\epsilon = -0.6$‰), pFe regeneration ($\epsilon = +0.15$‰), dFe scavenging from solution ($\epsilon = -0.3$‰) and dFe complexation

to natural organic ligands ($\epsilon = +0.6$‰) (Fig. 4, Supplementary Fig. 6 and supplementary information for extended discussion). The overall value for ε is considerably smaller than the expected value (between −2 and −3‰[23]) for biological reduction of Fe(III) to Fe(II) by cells, suggesting that dFe isotope fractionation is likely associated with the rapid trafficking of Fe between the dissolved pool and cells (Fig. 4). The recycling of Fe between pools may also amplify the $\delta^{56}$Fe$_{diss}$ signal. The model also produces $\delta^{56}$Fe$_{part}$ values within the range measured in the euphotic zone, supporting the idea that mixing and particle sinking are the primary mechanisms responsible for distributing the $\delta^{56}$Fe$_{diss}$ and $\delta^{56}$Fe$_{part}$ signals across the euphotic zone.

Iron bioavailability is known to influence the oxidation rates of ammonia and nitrite[24], such that both accumulate if Fe is limiting. An in-eddy minimum in $\delta^{56}$Fe$_{diss}$ (Fig. 4) coincided with peaks in heterotrophic bacterial cell numbers (Supplementary Fig. 3) and nitrite concentration at 150 m, and just below the peak in ammonium concentration (Supplementary Fig. 5). This minimum in $\delta^{56}$Fe$_{diss}$ and the maxima in nitrite, ammonia and heterotrophic bacterial cell numbers provides evidence for Fe control of the microbial recycling of organic matter in the mesopelagic. Note that no localised minimum in $\delta^{56}$Fe$_{diss}$ or peaks in heterotrophic bacterial cell abundance or ammonium and nitrite occurred below the euphotic zone at the SAZ or SOTS stations (Supplementary Figs. 3, 5) indicating that Fe limitation of the microbial component of the mesopelagic community is restricted to the eddy station (Supplementary Fig. 5). The model reveals seasonality in dFe and pFe concentrations suggesting that these signals of particulate Fe sinking and remineralization are initiated in spring and persist through to autumn (Supplementary Fig. 7), thus it is likely that their influence on Southern Ocean biogeochemistry may be even greater than suggested by the autumn observations presented here.

Satellite analysis of thousands of cyclonic eddies from the Southern Ocean reveals that they are generally cooler than surrounding waters but have higher chlorophyll concentrations in winter and spring and lower chlorophyll concentrations in summer and autumn[11,25]. These eddies have shallower mixed layers, particularly in winter and spring[25,26]. The increase in production during winter and spring likely results from an alleviation of light limitation[27], which is a key variable limiting plankton growth after winter mixing has reset the dFe concentration from levels that can limit phytoplankton production[16]. In contrast, shallow mixed layers during summer result in cyclonic eddies having a lower dFe inventory and hence low phytoplankton production compared to surrounding waters[25]. Our study confirms that cyclonic eddies can have extremely low dFe inventories, thus requiring the resident

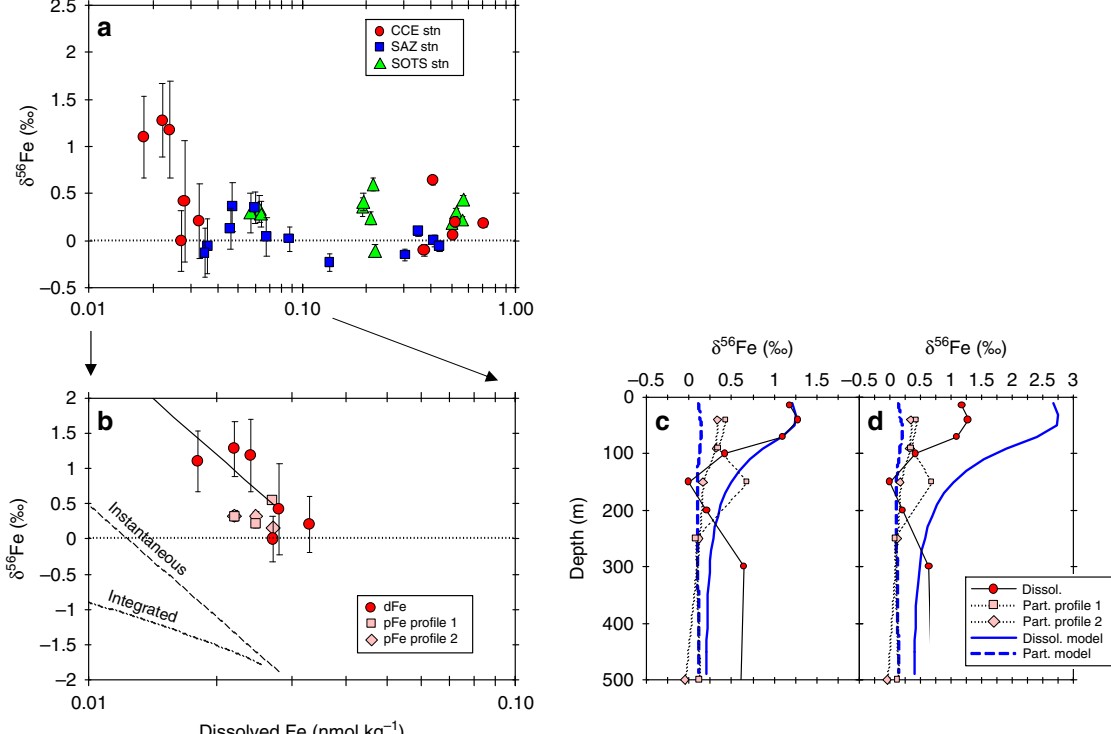

**Fig. 4 Iron isotope fractionation model calculations. a** Dissolved δ$^{56}$Fe values for samples collected at the CCE, SAZ, and SOTS stations versus Fe concentration. **b** Model curves for closed steady-state isotope fractionation (Eqs 2–4) of dFe for the CCE. The best fit $\varepsilon$ value for the cold core eddy dFe data is −2.3‰. One-D model profiles (blue lines) for dFe and pFe versus depth for **c**. $\varepsilon$ equal to −1.0‰ ($\alpha_{uptake}$ = 0.999) and **d**. $\varepsilon$ equal to −2.3 ‰ ($\alpha_{uptake}$ = 0.9977) along with profiles of dFe and pFe isotope composition versus depth for the CCE. Error bars for isotope measurements represent 2 s.e.m. Where error bars are not seen, they are within the size of the symbol. **c, d** Error bars have been removed for clarity.

plankton to upregulate iron uptake machinery and use recycling activities to sustain themselves. Such conditions favour small, non-siliceous cells, thus reducing export. At any time there are about 1200 eddies in the Southern Ocean, and approximately 50% of them are cyclonic[26], thus the extreme Fe limitation of Southern Ocean phytoplankton, as reported here, is likely to be the prevalent state during summer and autumn[25].

## Methods

**Voyage.** The cold-core cyclonic eddy was studied between 28 March and 3 April 2016 and was part of a GEOTRACES process study (Fig. 1). The eddy was about 190 km in diameter and was a stable feature that had formed approximately 1 month prior to sampling (Fig. 1, Supplementary Movie 1). It was formed by detaching from waters 2 degrees south near the Subantarctic Front[3,4]. Post formation, the eddy moved slowly northward across the northern extension of the subantarctic front (SAF-N; Supplementary Movie 1). As it moved, the eddy completed approximately 7 full rotations during is 109 day lifetime[3] The biogeochemical properties of the cold-core cyclonic eddy were contrasted with two other sites in the subantarctic zone. One was located at 51.90°S, 148.51°E, designated a Subantarctic zone site (SAZ) and the other at 46.77°S, 142.03°E, the Southern Ocean Time Series (SOTS) site (Fig. 1).

**CTD, nutrient sampling.** Conductivity, temperature, and depth (CTD) profile data and water samples for nutrients and biological parameters were collected with a winch-lowered package consisting of an SBE 911plus CTD, a Turner Designs fluorometer, and a 24-bottle SBE 32 Carousel water sampler. Salinities were calibrated to standard seawater (International Association for the Physical Sciences of the Ocean). Samples for phosphate, nitrate, nitrite, ammonia and silicic acid were collected and analysed at sea on unfiltered samples using a Seal AA3 segmented flow system following the procedures outlined by Armstrong et al.[28] and Wood et al.[29].

**Trace metal sampling.** Seawater samples for trace metal and isotope determination were collected using Teflon-coated, externally-sprung, 12-L Niskin bottles attached to an autonomous rosette equipped with a Conductivity Temperature Depth (CTD) unit (SeaBird 911 plus, USA). Upon retrieval, the Niskin bottles were transferred

into a clean container laboratory fitted with HEPA-filtered air workstations. Seawater samples for dissolved trace metal analysis were filtered through acid-cleaned 0.2-μm capsule filters (Supor AcroPak 200, Pall) and acidified with distilled nitric acid to a final pH ≤ 1.8. The sampling protocols followed recommendations in the GEOTRACES Cookbook (http://www.geotraces.org/science/intercalibration/222-sampling-and-sample-handling-protocols-for-geotraces-cruises).

Particulate trace metal samples were collected in situ onto acid-leached 0.2-μm Supor (142 mm diameter) filters (Pall, Australia) using six large-volume dual-head pumps (McLane Research Laboratories) deployed at various water depths. For most profiles, one pump depth was used as a blank check whereby only 4 L of water was pumped through the filter. This filter was then processed using the same procedure as the other samples.

**Primary productivity and iron uptake.** Net primary productivity and Fe uptake rates were determined for water column samples collected at six depths between 0 and 100 m[30]. Water samples were collected pre-dawn from trace metal clean Niskin bottles deployed on a trace metal rosette. Sampling depths were determined from in situ irradiance depth profiles obtained during midday CTD casts the day prior to collection. Samples were dispensed into 300 mL acid-washed polycarbonate bottles and spiked with 16 μCi of Sodium $^{14}$C-bicarbonate (NaH$^{14}$CO$_3$; specific activity 1.85 GBq mmol$^{-1}$; PerkinElmer) and 0.2 nmol L$^{-1}$ of an acidified $^{55}$Fe solution ($^{55}$FeCl$_3$ in 0.1 M Ultrapure HCl; specific activity 30 MBq mmol$^{-1}$; PerkinElmer). Six samples (five light and one dark bottle per irradiance) were incubated for 24 h in a deck-board incubator under natural sunlight at six light intensities (from 80 to 1.0% of incident irradiance). Light attenuation was adjusted by varying the layers of neutral density mesh and measured with a Biospherical Instruments QSL2101 Quantum Scalar PAR Sensor. The temperature of the incubator was controlled by a continuous supply of surface seawater.

Upon completion of the 24-h incubation, four replicate samples were serially vacuum-filtered (<10 mm Hg) through 20, 2.0, and 0.2 μm porosity polycarbonate filters (47 mm diameter; Poretics) separated by 200 μm nylon mesh. Two size-fractionated samples were washed with Titanium(III) EDTA—citrate reagent for 5 min to dissolve Fe (oxy)hydroxides and remove ferric ions bound to the outer membrane surface[31], and rinsed three times with 15 mL of 0.2 μm-filtered seawater[32,33] and the other two size-fractionated samples were rinsed only with 0.2 μm-filtered seawater. In addition, two samples were filtered through 0.2 μm filters (a total community light control, and a total community dark control). Data for the dark-corrected, size-fractionated samples rinsed with the Ti(III) EDTA—citrate reagent are reported here (i.e., intracellular Fe:C uptake ratios).

The filters were transferred to 20 mL scintillation vials (Wheaton) and acidified with 100 μL of 1.0 M HCl to volatilise any remaining inorganic carbon[30]. Samples were counted on a liquid scintillation counter (PerkinElmer Tri-Carb 2910 TR) with a dual-label counting protocol after the addition of 10 mL of liquid scintillation cocktail (UltimaGold, PerkinElmer). Unfiltered water samples (1 mL) were used to quantify the concentrations of added $^{14}C$ and $^{55}Fe$, and Fe:C uptake ratios for the size fractions were calculated from specific activities after accounting for both added and ambient dissolved Fe concentrations.

**Cell counts**. Flow cytometric analyses were performed following protocols outlined by Marie et al.[34]. Samples for cell counts were preserved with glutaraldehyde and stored frozen at −80 °C to protect against cell lysis and the loss of autofluorescence. Prior to analysis, frozen samples were rapidly thawed in a water bath at 70 °C for 3 min and aliquots were taken for autotrophic and prokaryote cell counts. Sample aliquots were kept on ice in the dark and promptly analysed on a Becton Dickinson FACScan flow cytometer fitted with a 488 nm laser. Milli-Q water was used as sheath fluid for all analyses. Before and after each run, samples were weighed to determine the amount of sample analysed.

Autotrophic cell abundance samples were prepared by pipetting 1 mL of sample to a clean 5 mL polycarbonate tube, with 2 μL of PeakFlow Green 2.5 μL beads (Invitrogen) added as an internal fluorescence and size standard. Each sample was run for 5 min at a high flow rate of 40 μL min$^{-1}$. Autotrophic cell populations were separated into regions based on their chlorophyll autofluorescence in red (FL3) versus orange (FL2) bivariate scatter plots. *Synechococcus* sp cells were determined from their high FL2 and low FL3 fluorescence. Pico- (<2 μm) and nanophytoplankton (2–20 μm) communities were determined from their relative cell size in side scatter (SSC) versus FL3 fluorescence bivariate scatter plots.

Samples for prokaryote cell abundance were prepared by pipetting 1 mL of sample to a clean 5 mL polycarbonate tube. Samples with high prokaryote cell counts were diluted to 1:10 with 0.2 μm filtered seawater (FSW) to remove underestimation of cell concentration from coincidence (100 μL sample in 900 μL FSW). Cells were stained for 20 min with 5 μL of SYBR Green I (Invitrogen) at a final dilution of 1:10,000. An additional 2 μL of PeakFlow Green 2.5 μL beads (Invitrogen) were added to the sample as an internal fluorescence and size standard. Each sample was run at a low flow rate of ~12 μL min$^{-1}$ for 3 min and prokaryote cell abundance was determined from bivariate scatter plots of SSC versus green (FL1) fluorescence.

**Iron isotope analysis**. Particulate samples for trace element and δ$^{56}$Fe determination were thawed and processed using previous acid digestion protocols[35–37]. For dFe isotope determination, seawater samples (2 L) were spiked with a $^{57}$Fe–$^{58}$Fe double spike[38,39]. Samples were left overnight to equilibrate, after which they were buffered to a pH of 4.5 with a trace-metal clean ammonium acetate buffer and then passed over 0.5 mL columns packed with Nobias PA Chelate PA1L resin (Hitachi-Hitec, Japan) at a flow rate of 2 mL min$^{-1}$. Samples were rinsed with 4 mL of ammonium acetate buffer solution (1% w w$^{-1}$) followed by elution with 4 mL of 1 mol L$^{-1}$ nitric acid. Samples were evaporated to dryness and redissolved with 0.5 mL of 6 mol L$^{-1}$ hydrochloric acid containing $H_2O_2$. Samples were further purified using an anion exchange procedure similar to that described by Poitrasson and Freydier[40]. Precleaning of the AG-MP1 resin involved rinsing with methanol, multiple washes with 6 mol L$^{-1}$ hydrochloric acid and 0.5 mol L$^{-1}$ nitric acid before storage in dilute nitric acid. When required ~200 μL columns filled with the precleaned anion exchange resin AG-MP1 (Bio-Rad), conditioned by washing with 0.5 mol L$^{-1}$ hydrochloric acid, 0.5 mol L$^{-1}$ nitric acid, Milli-Q water and finally 6 mol L$^{-1}$ hydrochloric acid before use. Between use, columns were stored filled in 2% w w$^{-1}$ nitric acid. Columns were typically used 5–8 times before being refilled with new precleaned resin. After sampling loading, salts and other elements not of interest were eluted from the column by passing 3 × 1 mL of 6 mol L$^{-1}$ hydrochloric acid. Iron was eluted with 3 × 1 mL of 0.5 mol L$^{-1}$ hydrochloric acid and evaporated to dryness. Samples were redissolved in either 0.30 or 0.35 mL of 2% (w w$^{-1}$) nitric acid. The blank associated with the anion exchange separation was 0.39 ± 0.34 ng (n = 4). Procedural concentration blanks for the whole process were determined by passing small volumes (~50 mL) of an in-house seawater standard with a concentration 0.78 ± 0.08 nmol kg$^{-1}$ over the Nobias PA Chelate PA1L resin and then through the whole elemental and Fe isotope separation procedure. The dissolved iron concentration for this smaller seawater was then scaled to 2 L thus allowing us to estimate the blank associated with the buffering of the sample, passing it over the Nobias PA Chelate PA1L resin and then over the anion exchange columns. The blank associated with this test was determined to be 0.40 ± 0.32 ng (n = 5). Note that we were not able to determine the isotope composition of the blank associated with the extraction and processing procedure, so the isotope values presented in have not been blank corrected. For dissolved samples, the total amount of Fe analysed ranged between 2 and 63 ng, thus the contribution of the blank to the lowest concentration samples could have been between as much 20 ± 17% of the lowest δ$^{56}$Fe$_{diss}$ signal, i.e., for samples collected from the upper water column within the CCE.

Iron isotopes were determined using a Neptune Plus multi-collector ICPMS (ThermoScientific) with an APEX-IR introduction system (ESI, USA) and with X-type skimmer cones. Samples were measured in high-resolution mode with $^{54}$Cr

interference correction on $^{54}$Fe and $^{58}$Ni interference correction on $^{58}$Fe. Iron isotope ratios ($^{56}$Fe/$^{54}$Fe) ratios are reported in delta notation (‰) relative to the IRMM-014 Fe isotope reference material (IRMM, Brussels) using the double spike ($^{57}$Fe–$^{58}$Fe) correction technique[38,39] where:

$$\delta^{56}Fe = \left( \frac{^{56}Fe/^{54}Fe_{sample}}{^{56}Fe/^{54}Fe_{IRMM-014}} - 1 \right) \times 1000, \quad (1)$$

The overall instrumental error for dFe and pFe samples ranged between ±0.04‰ and ±0.64‰ (2σ). For low concentration samples, the instrumental error increased with decreasing Fe concentration and was associated with instrumental noise (Supplementary Fig. 6)[41]. Multiple large volume (3 × 2 L) extractions and analysis of an in-house seawater standard had a reproducibility of 0.76 ± 0.07‰ (mean ± 2 standard deviation). Multiple analysis of a particulate sample had a reproducibility of 0.15 ± 0.06‰ (mean ± 2 standard deviation). Analysis of geological samples NOD-A-1 and BCR-2 produced values of −0.43 ± 0.03‰ and 0.04 ± 0.07‰ (mean ± 2 standard deviation), respectively, which were consistent with literature values of −0.42 ± 0.07[42] for NOD-A-1 and 0.03 ± 0.06[42] for BCR-2. The performance of the Fe isotope method was also assessed through an intercalibration exercise for samples from the GP13 and GP19 GEOTRACES campaigns at a crossover station located at 30°S; 170°W. The iron isotope results from the exercise were comparable—i.e., the trends seen in the GP13 and GP19 profiles are consistent with each other covering a dFe range between 0.017 and 0.72 nmol kg$^{-1}$ (Supplementary Fig. 8)[43].

Dissolved Fe concentration for each sample was calculated using sample weight and the amount double spike added to the sample. This calculation is based on isotope dilution using the known proportion of $^{58}$Fe in the $^{57}$Fe–$^{58}$Fe double spike[38,44]. Note that the dFe concentrations presented here were not blank corrected, thus, they represent an upper concentration bound.

As with all open ocean seawater work, during the collection and processing of samples contamination can hinder the production of accurate and meaningful data. The added challenge for Fe isotope studies, particularly for low concentration systems such as the Southern Ocean, is obtaining enough material for isotope analysis. For the result presented here, the dFe processing blank associated represents as much 20 ± 17% of the concentration and the isotope signal. While concentration uncertainties are highest for shallow samples collected in the CCE, the structure of the dFe concentration versus depth profile for this station, and indeed the other two stations, are oceanographically consistent, i.e., they have low surface water concentrations that increase with depth[45]. In a companion study, dissolved zinc concentration and zinc isotope results obtained from the same samples showed no indication of trace metal contamination associated with sample collection and processing[46]. For the dFe isotope results, there is also the added challenge of obtaining enough material for isotope analysis. Here we optimised the isotopic measurement of dFe by reducing the volume of each sample presented for analysis (0.3–0.35 mL) thereby upping its concentration to reduce errors associated with instrument noise[41,47]. We also utilised a spike-sample ratio ranging between 1 to 3 (spike $^{57}$Fe-$^{58}$Fe ratio = 1.05) such that measurement errors are minimised for $^{56}$Fe, $^{57}$Fe, and $^{58}$Fe. Even with these steps, the influence of instrument noise increased for low concentration Fe samples (Supplementary Fig. 9). While the uncertainty window around these measurements is larger than that for samples with a higher dFe concentration, the upper water column variations for δ$^{56}$Fe$_{diss}$ between 15 and 150 m are statistically distinct and oceanographically consistent. The enrichment of δ$^{56}$Fe$_{diss}$ within the euphotic zone is consistent with measurements made at 32.5°S, 150°W (Supplementary Fig. 8) and other recent measurements for low dFe concentration waters of the Southern Ocean[48]. Likewise, the decline in δ$^{56}$Fe$_{diss}$ values below the euphotic zone is consistent with measurements made at 32.5°S, 150°W (Supplementary Fig. 10), although one should be mindful that this station is outside of the Southern Ocean such that the biological community leading to variation in δ$^{56}$Fe$_{diss}$ is likely to be different.

**Iron isotope modelling**. The closed system equation for the isotopic evolution of dFe as it is consumed can be described as follows:

$$\delta^{56}Fe_{dissolved} = \delta^{56}Fe_{dissolved.100m} + \varepsilon \times \ln(f), \quad (2)$$

where ε represents the isotope enrichment factor between the product (biologically utilised Fe) and the substrate (dFe) and f esents the fraction of dFe relative to the concentration of dFe at 100 m. The evolution of the instantaneous or integrated isotope fractionation processes can be modelled using the following expressions:

$$\delta^{56}Fe_{particulate} = \delta^{56}Fe_{dissolved} - \varepsilon, \quad (3)$$

$$\delta^{56}Fe_{particulate} = \delta^{56}Fe_{dissolved} + \frac{\varepsilon \times \ln(f)}{1 - f}. \quad (4)$$

**1D biogeochemical modelling**. The potential processes that influence the distribution and isotope fractionation of dFe and pFe were explored using a 1D model (Supplementary Fig. 10). The rationale for using this 1D model is to explore the relative influence (and interplay) of processes such as phytoplankton utilisation of Fe, its complexation to natural organic ligands, its regeneration from sinking

organic matter and the role of scavenging on distribution and expression of isotope profiles. The model is based on Schlosser et al.[49] and includes one phytoplankton group and references key nutrients including nitrate, phosphate and Fe (Supplementary Fig. 7). The model includes mixing, which supplies nutrients into the euphotic zone and the main loss process for nutrients and Fe from the euphotic zone (organic matter export). The Fe component in the model also includes complexation to natural organic ligands, scavenging and the atmospheric supply of Fe through the deposition and dissolution of dust (Supplementary Information, Supplementary Fig. 10). The model also does not include advection, which we justify for several reasons: (i) vertical advection, i.e., upwelling, occurs in the Southern Ocean primarily south of the Polar Front and not in the SAZ and SAF regions examined here for the cold core eddy[50], (ii) latitudinal advection supplies waters with similar properties from upstream in the Antarctic circumpolar current[51], and can thus be ignored; and (iii) transport is dominated by northward Ekman transport, and while this does supply nutrients over the annual mean, in late summer surface concentrations between the SAF and PFZ are very uniform[52], so this term can also be neglected. The equations and values associated with each biogeochemical process are presented in the Supplementary Information and in Supplementary Tables 5, 6.

## Data availability

The main datasets supporting the findings of this study are available within this article and its Supplementary Information, and additional data are available from the corresponding author on request.

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

## Acknowledgements
This research was financially supported under Australian Research Council's Discovery programme (DP170102108; DP130100679) and ship time from Australia's Marine National Facility. We are grateful to the officers, crew, and research staff of the Marine National Facility and the R.V. Investigator for their help with sample collection and generation of hydrochemistry data. We are grateful to Stacy Deppeler for running the flow cytometry samples.

## Author contributions
M.J.E. and P.W.B. conceived the study with input from P.G.S. and T.W.T. on appropriate oceanographic sampling sites. M.J.E., R.F.S. and M.F. collected and analysed samples. M.J.E. wrote the paper with assistance from all co-authors.

## Competing interests
The authors declare no competing interests.
