## [Peer Review File · Nature Communications]

Reviewers' comments:

Reviewer #1 (Remarks to the Author):

In "Biogeochemical metrics reveal distinctive but unusual iron cycling in the Southern Ocean eddies" Ellwood et al discuss iron measurements taken in a Southern Ocean eddy. They find Fe cycling characteristics in the eddy, a cyclone, that are distinct from surrounding waters. They conclude that the isolation of waters due to the eddy's dynamics allow for those distinct characteristics to occur, and more generally, that these findings will help to better understand Southern Ocean eddy chl/biomass anomalies.

The study presents -to my knowledge- a unique assessment of natural iron cycling within a Southern Ocean eddy, comparing it to measurements taken in surrounding waters. The study contributes to explaining mechanisms causing biogeochemical anomalies of Southern Ocean eddies. Thus, it presents a valuable addition to the field. The paper is well written. I have only few comments (see below), mainly concerning referencing of previous literature and why the eddy features characteristic Fe cycling. I recommend the paper for publication.

Minor comments:

*Title: How about a shorter title, sth like "Distinct iron cycling in a Southern Ocean eddy"? Also, this is a case study about one eddy- that is, I suggest to stay with the singular in the title (see also comment below on how representative the eddy is for other cyclones).

*General: I understand that Fe cycling in the eddy is distinct compared to surrounding waters because the eddy is isolating waters; though, it is not entirely clear to me if the iron conditions of the eddy are typical for conditions south of the SAF where the eddy is originating from? That is, is "mere" advection of water, including its material properties, the main player in setting the distinct eddy iron characteristics (as suggested, e.g., in L70/71/76)? Or do the iron characteristics of the eddy further evolve during the ~1 month after its detachment from the SAF (as suggested, e.g., in L96)? Also, you highlight the high Fe-to-carbon ratio and efficient recycling/short residence times of Fe in the eddy - can you comment on the larger-scale potential biogeochemical and/or ecological implications?

*L20 "productivity ... low" and L28 "persistent productivity": Appears contradicting at first glance. I suggest to rephrase to make clearer. E.g., "even though low, sustained productivity"?

*L34/35 "a crucial role in...": Please provide references for this statement (e.g., references you use later on, see References comment below).

*L37 "typically have closed circulation leading to biogeochemical properties": I am somewhat hesitant with "typically" as the majority of eddies does not appear to be efficient in trapping waters, see also, e.g., Wang et al, 2015; the cyclone observed here rather appears to represent an extreme case in that it features a rather high temperature anomaly (see also comment below); can you comment on this?

*L46 "supply of Fe... usually via": Do you mean to say that the supply usually is provided via iron-enriched deeper waters? If so I suggest sth like "supply of Fe usually is provided via upwelling and upward mixing of deeper iron enriched waters..."; also, lateral advection of iron by eddies may play a role, too, may it not, see e.g., Xiu et al, 2011, as an example from the north Pacific?

*L59 "about 2C lower": This appears to be a distinct/intense eddy; see "typical"/mean SST anomalies of eddies, e.g., in Haumann et al, 2012, of (well) below 1C; the large temperature anomaly suggests that the eddy rather is an extreme case, likely special also in terms of its biogeochemistry, and not so much an eddy representative for most eddies (see also comment above); I am happy to be convinced otherwise, though. If possible, could you comment on how many of such distinct eddies occur in the region versus the number of weaker eddies with less pronounced physical/biogeochemical anomalies (e.g., based on satellite SLA and SST/Chl)?

*L77 "unique": Unique compared to what, e.g., "unique in this region" or so. Or delete the second part of the sentence - is it necessary here?

*Fig1: I suggest to add SSH or SLA contours, a typical proxy to identify eddies.

Referencing, a few suggestions:

- *L34, ref 2: Include the observational paper Chelton et al, 2011, instead of the modeling work by Thomson et al 2010 (which focuses on fronts rather than eddies)?
- *L35: In my comment above, I ask to provide references here, you could include, e.g. the references you use later, 7, 8, Sheen et al 2014 & Pollard et al, 2006
- *L39, refs 1, 7, 8: Remove Frenger et al, 2015, Pollard et al, 2006, Sheen et al, 2014 (possibly include Sheen/Pollard above, see previous comment) and rather use observational papers that highlight local biogeochemical anomalies of eddies e.g., Lehahn et al, 2011.
- *L41, refs 8: Remove Sheen et al, 2014, possibly include Ansorge et al, 2010
- *L48, ref 1: Include paper with the original idea, Flierl, 1981
- *L54ff (paragraph): Here you introduce the observations of the eddy; please include references early on in this paragraph that discuss the same eddy and use the same observations, e.g., ref 13/23, Moreau et al, 2017 and Patel et al, 2019.
- *L163-167: I suggest to add a sentence here, embedding these conclusions/hypotheses on light/iron limitation in eddies in previous works; do the statements/hypotheses (dis)agree with findings/hypotheses, e.g., of the observational works of Dawson et al, 2018, Frenger et al, 2018, and the modeling work of Song et al, 2018? Also, Moreau et al, 2017, already mention low iron concentrations in this eddy and discuss what (iron/light/grazing....) may limit productivity in the eddy. Maybe this is just about making overly clear what main points you would like to convey with your paper.

References:

- *Ansorge et al., 2010, Polar, Biology, <https://link.springer.com/article/10.1007%2Fs00300-009-0752-9>
- *Chelton et al, 2011, Progr in Oceanography, <https://doi.org/10.1016/j.pocean.2011.01.002>
- *Flierl, 1981, Geophysical & Astrophysical Fluid Dynamics, <https://doi.org/10.1080/03091928108208773>
- *Frenger et al, 2018, Biogeosciences, <https://www.biogeosciences.net/15/4781/2018/>
- *Hausmann et al, 2012, The observed signature of mesoscale eddies in sea surface temperature and the associated heat transport, DSR, <https://linkinghub.elsevier.com/retrieve/pii/S0967063712001720>
- *Kahru et al., 2007, GRL, <https://agupubs.onlinelibrary.wiley.com/doi/full/10.1029/2007GL030430>
- *Lehahn et al, 2011, Long range transport of a quasi isolated chlorophyll patch by an Agulhas ring, GRL, <https://agupubs.onlinelibrary.wiley.com/doi/full/10.1029/2011GL048588>
- *Moreau et al, 2017, GBC, <https://agupubs.onlinelibrary.wiley.com/doi/full/10.1002/2017GB005669>
- *Patel et al, 2019, JGR, <https://agupubs.onlinelibrary.wiley.com/doi/full/10.1029/2018JC014655>
- *Song et al, 2018, GRL, <https://agupubs.onlinelibrary.wiley.com/doi/full/10.1029/2017GL076246>
- *Wang et al, 2015, GRL, <https://agupubs.onlinelibrary.wiley.com/doi/full/10.1002/2015GL064089>
- *Xiu et al, 2011, Iron flux induced by Haida eddies in the Gulf of Alaska, GRL, <https://agupubs.onlinelibrary.wiley.com/doi/full/10.1029/2011GL047946>

Reviewer #2 (Remarks to the Author):

This manuscript presents new data on Fe concentrations and Fe isotopes in the Southern Ocean, and interprets these data in the context of a 1D model of isotope cycling and other biogeochemical data.

The manuscript does not present enough information about methods to judge whether the underlying data are correct. This is particularly worrisome because the information which the authors do present hints that some data may be incorrect. I urge the authors to engage in significant additional methods testing and/or present the results of any significant methods testing which has already been performed. My three most significant concerns regarding methods are below:

1) Blank for Fe isotopes. The authors state a methods blank of 0.4 ng for Fe isotopes, yet they use 200 uL of AG-MP1 resin for purification. AG-MP1 resin has quite a high resin blank, and multiple publications suggest resin blanks for 200 uL resin would be roughly an order of magnitude higher (e.g. Dauphas et al. Anal Chem 2004, 1 mL resin, 20 ng Fe blank; Conway et al Anal Chim Acta, 2013, 20 uL resin 0.3 ng Fe blank; and others). The process used here for determining blanks was not clearly explained, aside from a sentence about running a 50 mL sample, and then calculating "based on a 2 L sample". Wouldn't it have been more straightforward just to report the blank obtained for the 50 mL sample, as this represents at least a minimum blank for a 2L sample? In any case, if the authors are really able to provide a 10-fold reduction in blank compared to all previous studies using the same anion exchange resin, this should be highlighted and clearly described.

Even if the reported blank of 0.4 ng is correct, more information needs to be provided. For the lowest concentration samples, the blank would constitute about 30% of the entire Fe in the samples. Was any sort of blank correction done? In order to do such a blank correction, the authors would need to know not only the concentration of the blank (and its variability) but also its isotopic composition (and its variability). How was this determined?

2) Fe uptake rates. Little information was provided about how Fe uptake rates were measured except that they were determined using ^{55}Fe . However, such experiments are notoriously difficult to perform. Specifically, it is quite common in such experiments that some of the "uptake" just represents surface adsorption or precipitation, rather than true biological internalization. This would invalidate the author's conclusion about the recycling timescale of Fe being ~ 1 day. Was the added Fe bound to any sort of ligand? Do the authors have any way of knowing whether the Fe was actually translocated across the cell membrane, or just precipitated on the cell surface?

3) Fe concentrations. Surprisingly for a manuscript focused so heavily on Fe concentrations, the authors appear not to have described the methods they used to measure Fe concentrations. Typical blanks for Fe concentration by Seafast are 150 pM. Blanks for Fe concentration analysis based on large-volume stable isotope samples may be smaller, but they would suffer from the same lack of clear description as described for $\delta^{56}\text{Fe}$ in point 1. What sort of ICPMS instrument was used to measure Fe concentrations? What was the blank for this method? How was blank and blank variability determined?

Reviewer #3 (Remarks to the Author):

In this paper the authors present an analysis of the Fe chemistry of a Southern Ocean cold-core eddy. The analysis of Fe speciation and uptake rates is shown to infer that, while biomass and productivity in the eddy is low, the Fe uptake rate is high, suggesting rapid turnover of the Fe pool. The control of Fe availability on production in the S. Ocean has been extensively studied and this paper focusing on the role of mesoscale eddies adds to this story. The paper is well written and the data presented seems robust.

I felt however that the message of the paper was not well stated. The abstract suggests that a main finding is that fast Fe turnover sustains 'persistent production' even when Fe is low; however, the concluding remarks suggest that the main message is 'extreme Fe limitation is prevalent in summer and autumn'. This latter argument seems weaker than that suggested in the abstract (as we know Fe limits production in the S. Ocean). Can the authors relate the findings more to carbon and state/quantify the increase in production (albeit it low) as a result of fast recycling? This would strengthen the findings of the paper.

Minor comments:

A legend (similar to that on Figure 2) on figure 3 would help.

We thank the three reviewers for their constructive comments that have helped to improve the manuscript. Our responses to reviewer comments are highlighted in blue.

Reviewers' comments:

Reviewer #1 (Remarks to the Author):

In “Biogeochemical metrics reveal distinctive but unusual iron cycling in the Southern Ocean eddies” Ellwood et al discuss iron measurements taken in a Southern Ocean eddy. They find Fe cycling characteristics in the eddy, a cyclone, that are distinct from surrounding waters. They conclude that the isolation of waters due to the eddy's dynamics allow for those distinct characteristics to occur, and more generally, that these finding will help to better understand Southern Ocean eddy chl/biomass anomalies.

The study presents -to my knowledge- a unique assessment of natural iron cycling within a Southern Ocean eddy, comparing it to measurements taken in surrounding waters. The study contributes to explaining mechanisms causing biogeochemical anomalies of Southern Ocean eddies. Thus, it presents a valuable addition to the field. The paper is well written. I have only few comments (see below), mainly concerning referencing of previous literature and why the eddy features characteristic Fe cycling. I recommend the paper for publication.

Thank you for your positive and constructive comments.

Minor comments:

*Title: How about a shorter the title, sth like “Distinct iron cycling in a Southern Ocean eddy”? Also, this is a case study about one eddy- that is, I suggest to stay with the singular in the title (see also comment below on how representative the eddy is for other cyclones).

Amended. See line 1

*General: I understand that Fe cycling in the eddy is distinct compared to surrounding waters because the eddy is isolating waters; though, it is not entirely clear to me if the iron conditions of the eddy are typical for conditions south of the SAF where the eddy is originating from? That is, is “mere” advection of water, including its material properties, the main player in setting the distinct eddy iron characteristics (as suggested, e.g., in L70/71/76)? Or do the iron characteristics of the eddy further evolve during the ~1 month after it's detachment from the SAF (as suggested, e.g., in L96)? Also, you highlight the high Fe-to-carbon ratio and efficient recycling/short residence times of Fe in the eddy – can you comment on the larger-scale potential biogeochemical and/or ecological implications?

The dFe characteristics of the eddy do appear to have evolved as the eddy has aged. We have explored with the macronutrient concentrations which are slightly lower than for the waters where eddy developed. This is mentioned on lines 79 to 86.

We have included a sentence in the at the main text highlighting that eddy conditions would favor small cells and thus reduce carbon export. See lines 179 to 182.

*L20 “productivity ... low” and L28 “persistent productivity”: Appears contracting at first glance. I suggest to rephrase to make clearer. E.g., “, even though low, sustained productivity”?

Rephrased to “with cells upregulating iron uptake and using recycling processes to sustain themselves” see line 28

*L34/35 “a crucial role in...”: Please provide references for this statement (e.g., references you use later on, see References comment below).

Two references have been added; Patel et al. (2019) and Patel et al. (2019). See line 35

*L37 “typically have closed circulation leading to biogeochemical properties”: I am somewhat hesitant with “typically” as the majority of eddies does not appear to be efficient in trapping waters, see also, e.g., Wang et al, 2015; the cyclone observed here rather appears to represent an extreme case in that it features a rather high temperature anomaly (see also comment below); can you comment on this?

The system that we studied is akin to what Frenger et al. (2018) classified as a trapping or monopole eddy. We have rephrased lines section to mention this in the text. See line 37

*L46 “supply of Fe... usually via”: Do you mean to say that the supply usually is provided via iron-enriched deeper waters? If so I suggest sth like “supply of Fe usually is provided via upwelling and upward mixing of deeper iron enriched waters...”; also, lateral advection of iron by eddies may play a role, too, may it not, see e.g., Xiu et al, 2011, as an example from the north Pacific?

Eddies may play a role in nutrient and iron vertical and lateral supply depending on where they form. The warm core Haida eddy referred to in the Xiu et al. (2011) study transported warmer, fresher, and nutrient- and iron-enriched water offshore from the British Columbian coast. For our cold core eddy we observed elevated nutrient levels relative to the surrounding waters, but these levels are lower than those in the waters where the eddy likely formed. The consumption of nutrients within the eddy during its transit may well explain why dissolved iron is lower than in the surrounding waters. We have mentioned nutrient consumption on lines 79 to 86

*L59 “about 2C lower”: This appears to be a distinct/intense eddy; see “typical”/mean SST anomalies of eddies, e.g., in Haumann et al, 2012, of (well) below 1C; the large temperature anomaly suggests that the eddy rather is an extreme case, likely special also in terms of its biogeochemistry, and not so much an eddy representative for most eddies (see also comment above); I am happy to be convinced otherwise, though. If possible, could you comment on how many of such distinct eddies

occur in the region versus the number of weaker eddies with less pronounced physical/biogeochemical anomalies (e.g., based on satellite SLA and SST/Chl)?

We have amended to the text to account for this point. Our was typical in size and life span but was more intense in terms of rotation speed and temperature gradients with respect surrounding waters. See lines 58-64.

*L77 “unique”: Unique compared to what, e.g., “unique in this region” or so. Or delete the second part of the sentence – is it necessary here?

Corrected.

*Fig1: I suggest to add SSH or SLA contours, a typical proxy to identify eddies.

We have included an extra panel to Figure 1 showing the sea level anomaly (SLA) for the region. The panel highlights that the eddy is characterised by lower SLA compared to surrounding waters. For decided to leave in the temperature contours for panels 1b and 1c as the nice define the edge of the eddy.

Referencing, a few suggestions:

*L34, ref 2: Include the observational paper Chelton et al, 2011, instead of the modeling work by Thomson et al 2010 (which focuses on fronts rather than eddies)?

Done.

*L35: In my comment above, I ask to provide references here, you could include, e.g. the references you use later, 7, 8, Sheen et al 2014 & Pollard et al, 2006

Amended so this now include (Moreau et al., 2017; Patel et al., 2019; Pollard et al., 2006; Sheen et al., 2014). Line 35

*L39, refs 1, 7, 8: Remove Frenger et al, 2015, Pollard et al, 2006, Sheen et al, 2014 (possibly include Sheen/Pollard above, see pervious comment) and rather use observational papers that highlight local biogeochemical anomalies of eddies e.g., Lehahn et al, 2011.

Done. Line 40

*L41, refs 8: Remove Sheen et al, 2014, possibly include Ansorge et al, 2010

Amended. Line 41

*L48, ref 1: Include paper with the original idea, Flierl, 1981

Done. line 48

*L54ff (paragraph): Here you introduce the observations of the eddy; please include references early on in this paragraph that discuss the same eddy and use the same observations, e.g., ref 13/23, Moreau et al, 2017 and Patel et al, 2019.

Amended. Line 55

*L163-167: I suggest to add a sentence here, embedding these conclusions/hypotheses on light/iron limitation in eddies in previous works; do the statements/hypotheses (dis)agree with findings/hypotheses, e.g., of the observational works of Dawson et al, 2018, Frenger et al, 2018, and the modeling work of Song et al, 2018? Also, Moreau et al, 2017, already mention low iron concentrations in this eddy and discuss what (iron/light/grazing....) may limit productivity in the eddy.

Done see lines 174 to 177

Maybe this is just about making overly clear what main points you would like to convey with your paper.

References:

- *Ansorge et al., 2010, Polar, Biology, <https://link.springer.com/article/10.1007%2Fs00300-009-0752-9>
- *Chelton et al, 2011, Progr in Oceanography, <https://doi.org/10.1016/j.pocean.2011.01.002>
- *Flierl, 1981, Geophysical & Astrophysical Fluid Dynamics, <https://doi.org/10.1080/03091928108208773>
- *Frenger et al, 2018, Biogeosciences, <https://www.biogeosciences.net/15/4781/2018/>
- *Hausmann et al, 2012, The observed signature of mesoscale eddies in sea surface temperature and the associated heat transport, DSR, <https://linkinghub.elsevier.com/retrieve/pii/S0967063712001720>
- *Kahru et al., 2007, GRL, <https://agupubs.onlinelibrary.wiley.com/doi/full/10.1029/2007GL030430>
- *Lehahn et al, 2011, Long range transport of a quasi isolated chlorophyll patch by an Agulhas ring, GRL, <https://agupubs.onlinelibrary.wiley.com/doi/full/10.1029/2011GL048588>
- *Moreau et al, 2017, GBC, <https://agupubs.onlinelibrary.wiley.com/doi/full/10.1002/2017GB005669>
- *Patel et al, 2019, JGR, <https://agupubs.onlinelibrary.wiley.com/doi/full/10.1029/2018JC014655>
- *Song et al, 2018, GRL, <https://agupubs.onlinelibrary.wiley.com/doi/full/10.1029/2017GL076246>
- *Wang et al, 2015, GRL, <https://agupubs.onlinelibrary.wiley.com/doi/full/10.1002/2015GL064089>
- *Xiu et al, 2011, Iron flux induced by Haida eddies in the Gulf of Alaska, GRL, <https://agupubs.onlinelibrary.wiley.com/doi/full/10.1029/2011GL047946>

Reviewer #2 (Remarks to the Author):

This manuscript presents new data on Fe concentrations and Fe isotopes in the Southern Ocean, and interprets these data in the context of a 1D model of isotope cycling and other biogeochemical data.

The manuscript does not present enough information about methods to judge whether the

underlying data are correct. This is particularly worrisome because the information which the authors do present hints that some data may be incorrect. I urge the authors to engage in significant additional methods testing and/or present the results of any significant methods testing which has already been performed. My three most significant concerns regarding methods are below:

Thank you for your constructive comments. We have addressed your concerns below

1) Blank for Fe isotopes. The authors state a methods blank of 0.4 ng for Fe isotopes, yet they use 200 uL of AG-MP1 resin for purification. AG-MP1 resin has quite a high resin blank, and multiple publications suggest resin blanks for 200 uL resin would be roughly an order of magnitude higher (e.g. Dauphas et al. Anal Chem 2004, 1 mL resin, 20 ng Fe blank; Conway et al Anal Chim Acta, 2013, 20 uL resin 0.3 ng Fe blank; and others). The process used here for determining blanks was not clearly explained, aside from a sentence about running a 50 mL sample, and then calculating "based on a 2 L sample". Wouldn't it have been more straightforward just to report the blank obtained for the 50 mL sample, as this represents at least a minimum blank for a 2L sample? In any case, if the authors are really able to provide a 10-fold reduction in blank compared to all previous studies using the same anion exchange resin, this should be highlighted and clearly described.

We have increased the detail within the relevant Section, and have spelled out the blanks we obtained for the anion-exchange separation component of the method, and for the full iron extraction and separation process. For these we get blanks of 0.39 ± 0.34 ng ($n = 4$) and 0.40 ± 0.32 ng ($n = 5$), respectively. We have also detailed how we precleaned our AG-MP1 resin and how the columns were stored between use. At face value, most of the blank can be ascribed to the anion exchange process. See lines 227 to 306

With respect to the earlier work of Dauphas et al. (2004), their blank results were for the AG1-X8 and AG50W-X4 resins, not the AG-MP1 resin.

Even if the reported blank of 0.4 ng is correct, more information needs to be provided. For the lowest concentration samples, the blank would constitute about 30% of the entire Fe in the samples. Was any sort of blank correction done? In order to do such a blank correction, the authors would need to know not only the concentration of the blank (and its variability) but also its isotopic composition (and its variability). How was this determined?

We have included more details about how we dealt with blank contribution within samples. Because we were not able to determine the iron isotope composition of the blank we chose not to correct the results. For all our results, a blank correction of the isotope data would have resulted in values that are statistically indistinguishable from the results presented. For example, for the 70m sample (2 ng of Fe) from the cold core eddy, we obtained a dissolved Fe isotope value of 1.10 ± 0.43 ‰ (2.SE). Correcting this 70m value with a 20% blank contribution (and assuming a lithogenic isotope value for the blank of 0.1‰) produces a result of 1.35 ± 0.43 ‰. The difference between this and the measured value is not statistically significant. Likewise, if we take the dissolved iron

isotope value for the 300 m depth sample (total of 45 ng Fe) from the cold core eddy, we obtain a blank corrected isotope value of $0.64 \pm 0.05 \%$ which is indistinguishable from its measured isotope value of $0.64 \pm 0.05 \%$.

Because we chose not to correct our iron isotope results we also chose not correct our concentration results. For the majority of the samples, this had no significant bearing on the final results. For a few this correction will be significant and thus they represent upper concentration bound.

See lines 301-306 and 326-327

2)Fe uptake rates. Little information was provided about how Fe uptake rates were measured except that they were determined using ^{55}Fe . However, such experiments are notoriously difficult to perform. Specifically, it is quite common in such experiments that some of the “uptake” just represents surface adsorption or precipitation, rather than true biological internalization. This would invalidate the author’s conclusion about the recycling timescale of Fe being ~ 1 day. Was the added Fe bound to any sort of ligand? Do the authors have any way of knowing whether the Fe was actually translocated across the cell membrane, or just precipitated on the cell surface?

Earlier field studies used ^{55}Fe concentrations one to two orders of magnitude higher (i.e., $2 - 20 \text{ nmol L}^{-1}$) than we used in this study (Maldonado et al. 2005, Strzepek et al. 2005, McKay et al. 2005, Hopkinson et al. 2012). In those cases, the addition of Fe bound to EDTA (usually $10 \mu\text{M}$) was warranted. However, we added a much lower concentration of ^{55}Fe in our experiments: 0.2 nmol L^{-1} of ^{55}Fe was added to each sample bottle as an acidified $^{55}\text{FeCl}_3$ solution (0.1 M Quartz-distilled HCl). No EDTA was used to complex the ^{55}Fe as the concentration of ^{55}Fe added is below the empirically observed threshold for precipitation of Fe (oxy)hydroxides ($\sim 0.7 \text{ nmol L}^{-1}$; Sunda and Huntsman 1995).

Surface adsorption or precipitation, rather than true biological internalization: The Ti(III) EDTA-citrate reagent that we used in our experiments was developed 30 years ago to remove extracellularly bound Fe from phytoplankton cells. The use of the Ti(III) EDTA-citrate reagent has been shown to be exceptionally successful in removing both freshly formed and aged particulate abiotic ferric oxyhydroxides ($> 99\%$ Fe removal; Hudson and Morel 1989, Tang and Morel 2006, Hassler and Schoemann 2009). The results obtained in the original paper describing the method (Hudson and Morel 1989) have been verified several times for both lab and field samples (Twining et al. 2004, Tang and Morel 2006, Hassler and Schoemann 2009). The Ti(III) reagent has shown to be an efficient washing agent as it relies on both redox potential and strong binding affinity for Fe(III) to first reduce ^{55}Fe oxyhydroxides that may precipitate on the cell surface, and then subsequently chelates the ^{55}Fe liberated

into the dissolved phase. The extracellular ^{55}Fe is then removed from the filters with multiple seawater rinses.

Combining data from all filter fractions and locations for the present study, Fe uptake rates and Fe:C uptake ratios were 2.65 ± 0.14 ($\pm\text{SE}$, $n = 54$) times higher, and C uptake rates 1.07 ± 0.03 times higher in unwashed samples compared to Ti(III) reagent-washed samples, indicating that: 1) intracellular Fe accounted for ~38% of total Fe 'uptake'; and 2) the Ti(III) EDTA - citrate reagent did not damage cells appreciably, as indicated by the good agreement in the C uptake rates between washed and unwashed filters.

We note also that the authors have a proven, >15 year, track record conducting these dual-label radiotracer experiments in the field, and have published extensively on how ^{55}Fe concentrations, ^{55}Fe chelation, and the use of the Ti reagent influence Fe:C uptake rates (e.g. Maldonado et al. 2005, Strzepek et al. 2005, McKay et al. 2005).

When a low concentration of ^{55}Fe is used (e.g. 0.2 nmol L^{-1}) and the cells are then washed with the Ti(III) reagent, Fe quotas measured with the radiotracer method are in good agreement with those obtained from single-cell measures of Fe content using synchrotron x-ray fluorescence (SXRF; Twining et al. 2004, Strzepek et al. 2005).

See lines 223 to 251

3)Fe concentrations. Surprisingly for a manuscript focused so heavily on Fe concentrations, the authors appear not to have described the methods they used to measure Fe concentrations. Typical blanks for Fe concentration by Seafast are 150 pM. Blanks for Fe concentration analysis based on large-volume stable isotope samples may be smaller, but they would suffer from the same lack of clear description as described for $\delta^{56}\text{Fe}$ in point 1. What sort of ICPMS instrument was used to measure Fe concentrations? What was the blank for this method? How was blank and blank variability determined?

In the Methods section, we now present the blanks in two forms: the blank associated with anion-exchange separation component of the method, and for the full iron extraction and separation from each sample. To determine, the dissolved iron concentration we made use of the amount of double spike used to determine the iron isotope composition of each sample. Specifically, we state that "Dissolved iron concentration for each sample is calculated from sample volume and the amount double spike added to the sample. This calculation is based on isotope dilution using the known proportion of ^{58}Fe in the ^{57}Fe - ^{58}Fe double spike. Note that the dFe concentrations presented here were not blank corrected, thus, they represent an upper concentration bound." The instrument used to make iron isotope measurements was a ThermoScientific NeptunePlus which is mentioned in the methods section.

See lines 324 to 327

Reviewer #3 (Remarks to the Author):

In this paper the authors present an analysis of the Fe chemistry of a Southern Ocean cold-core eddy. The analysis of Fe speciation and uptake rates is shown to infer that, while biomass and productivity in the eddy is low, the Fe uptake rate is high, suggesting rapid turnover of the Fe pool. The control of Fe availability on production in the S. Ocean has been extensively studied and this paper focusing on the role of mesoscale eddies adds to this story. The paper is well written and the data presented seems robust.

Thank you for your positive and constructive comments.

I felt however that the message of the paper was not well stated. The abstract suggests that a main finding is that fast Fe turnover sustains 'persistent production' even when Fe is low; however, the concluding remarks suggest that the main message is 'extreme Fe limitation is prevalent in summer and autumn'. This latter argument seems weaker than that suggested in the abstract (as we know Fe limits production in the S. Ocean). Can the authors relate the findings more to carbon and state/quantify the increase in production (albeit it low) as a result of fast recycling? This would strengthen the findings of the paper.

We have reworded the last sentence in question in the Abstract, and also the last paragraph of the main text. See lines 26 to 30 and 179 to 184

Minor comments:

A legend (similar to that on Figure 2) on figure 3 would help.

Done. - we have station abbreviations on these panels

References

Dauphas, N., Janney, P.E., Mendybaev, R.A., Wadhwa, M., Richter, F.M., Davis, A.M., van Zuilen, M., Hines, R., Foley, C.N. (2004). Chromatographic Separation and Multicollection-ICPMS Analysis of Iron. Investigating Mass-Dependent and -Independent Isotope Effects. *Anal. Chem.* 76, 5855-5863.

- Frenger, I., Münnich, M., Gruber, N. (2018). Imprint of Southern Ocean mesoscale eddies on chlorophyll. *Biogeosciences* 15, 4781-4798.
- Moreau, S., Penna, A.D., Llorc, J., Patel, R., Langlais, C., Boyd, P.W., Matear, R.J., Phillips, H.E., Trull, T.W., Tilbrook, B., Lenton, A., Strutton, P.G. (2017). Eddy-induced carbon transport across the Antarctic Circumpolar Current. *Global Biogeochem. Cycles* 31, 2017GB005669.
- Patel, R.S., Phillips, H.E., Strutton, P.G., Lenton, A., Llorc, J. (2019). Meridional Heat and Salt Transport Across the Subantarctic Front by Cold-Core Eddies. *J. Geophys. Res. Oceans* 124, 981-1004.
- Pollard, R., Tréguer, P., Read, J. (2006). Quantifying nutrient supply to the Southern Ocean. *J. Geophys. Res. Oceans* 111, C05011.
- Sheen, K.L., Naveira Garabato, A.C., Brearley, J.A., Meredith, M.P., Polzin, K.L., Smeed, D.A., Forryan, A., King, B.A., Sallee, J.B., St. Laurent, L., Thurnherr, A.M., Toole, J.M., Waterman, S.N., Watson, A.J. (2014). Eddy-induced variability in Southern Ocean abyssal mixing on climatic timescales. *Nat. Geosci.* 7, 577-582.
- Xiu, P., Palacz, A.P., Chai, F., Roy, E.G., Wells, M.L. (2011). Iron flux induced by Haida eddies in the Gulf of Alaska. *Geophys. Res. Lett.* 38.

Distinct iron cycling in a Southern Ocean eddy

Michael J. Ellwood*¹, Robert F. Strzepek², Peter G. Strutton^{2,3}, Thomas W. Trull⁴, Marion Fourquez²
and Philip W. Boyd².

1. Research School of Earth Sciences, Australian National University, Canberra, Australia

2. Institute for Marine and Antarctic Studies, University of Tasmania, Hobart, Australia

3. Australian Research Council Centre of Excellence for Climate Extremes, University of Tasmania,
Hobart, Australia

4. CSIRO Oceans and Atmosphere, Hobart, Australia

11October19 version

Style Definition: Default Paragraph Font

Deleted: Biogeochemical metrics reveal distinctive but unusual...

Deleted: eddies

Deleted: 30July19

**Abstract**

Mesoscale eddies are ubiquitous in the ocean, controlling ocean-atmosphere heat and gas
exchange. However, their physical influence on phytoplankton production in the iron-limited
Southern Ocean remains unknown. Here, we probed Fe cycling in a subantarctic cold-core eddy
during the austral autumn. In-eddy, surface dissolved Fe concentrations and phytoplankton
productivity and biomass were exceedingly low relative of external waters. In contrast, in-eddy
phytoplankton Fe-to-carbon uptake ratios were elevated 2-6 fold, indicating upregulated
intracellular Fe acquisition, thus reducing the residence time of Fe in the dissolved pool to ~1
27 day. Dissolved Fe isotope values were isotopically heavy in the euphotic zone, indicative of
28 extensive trafficking of Fe by cells within the eddy. Below the euphotic zone, Fe isotope values
were lighter and coincident with peaks in recycled nutrients and cell abundance, again indicative
of enhanced microbially-mediated Fe recycling. Our measurements show that the isolated
nature of eddies can produce distinctly different Fe biogeochemistry compared to surrounding
waters with cells upregulating iron uptake and using recycling processes to sustain themselves.
Recognising these eddy properties is essential to understanding how they contribute to the
biophysicochemical structure of the Southern Ocean.

Words = 186

**Main text**

Mesoscale eddies are ubiquitous in the ocean^{1,2} and play a crucial role in the transfer of heat, carbon
and nutrients between the deeper ocean, surface waters and the atmosphere³⁻⁶. Cold-core eddies in
the Southern Ocean are defined by strong clockwise rotation, cooler temperatures and negative sea-
surface height anomalies^{7,8}. These eddies can have closed circulation thus 'trapping' ^{ref9} the
biogeochemical properties of these water such as nutrients, chlorophyll and particle concentrations
differ compared to external waters^{2,9-11}. They can transport these biogeochemical properties vast
distances¹² thus they are important from an oceanographic point of view, especially if they cross
water mass boundaries such as the Polar Front or the Subantarctic Front^{7,11,13}.

The concentration of dissolved Fe (dFe) in remote Southern Ocean surface waters, away from
continental and island input sources, is typically sub-nanomolar (60-200 pmol kg⁻¹) ^{ref14,15}. The lower
limit for this dFe range is thought to be controlled by organic complexation and atmospheric supply.
In the Southern Ocean, atmospheric inputs are very low and the supply of Fe usually is provided via
upwelling and upward mixing of deeper iron-enriched waters^{14,16}. However, eddies can become

Deleted: microbial

Deleted: , including persistent production even when Fe is strongly depleted but efficiently recycled in surface waters.

Deleted: 189, 190 max

Deleted: deep

Deleted: .

Deleted: ^{3,4}. These eddies typically have closed circulation leading to biogeochemical properties such as nutrient, chlorophyll and particle concentrations that differ from external 
[revised manuscript text omitted]

Deleted: suggest that dFe concentrations should be

Deleted: unique

Deleted: the

Deleted: ¹⁰.

Deleted:) ¹⁴.

Deleted: ¹⁵.

Deleted: was

Deleted: ^{11,16}.

Deleted: ¹⁷

inventory of 1.87 $\mu\text{mol Fe m}^{-2}$ for the euphotic zone, we estimate a residence time of 1 day for dFe.
This short residence time indicates that Fe is being heavily trafficked within the euphotic zone
between the dissolved pool and the microbial community. The increased importance of iron
recycling favours smaller phytoplankton cells, which is reflected in the cell abundances, the size-
fractionated iron uptake and the Fe:C ratio datasets (Figures S3 and S4).

The elevated in-eddy Fe:C uptake ratios also raise the question as to how phytoplankton are
enhancing dFe uptake. Enhanced dFe uptake can occur through a combination of processes¹⁴,
including increased production of Fe transporters on the surface of cells, a reduction in cell size, the
production of Fe binding ligands, and the use of Fe(III) reductase proteins to enhance Fe(II)
production and hence the acquisition of Fe from organic complexes. We used the isotopic
composition of dFe and pFe to probe Fe uptake within the eddy.

The isotope composition of dFe ($\delta^{56}\text{Fe}_{\text{diss}}$) showed distinct variability with depth and between
stations (Figure 3). In the euphotic zone (0-85m) of the cold core eddy, $\delta^{56}\text{Fe}_{\text{diss}}$ values are
isotopically heavy at +1.19 ‰ and distinct ($p < 0.005$, T-test) to that of pFe ($\delta^{56}\text{Fe}_{\text{part}}$) and the \$\delta^{56}\text{Fe}_{\text{diss}}\$
composition for waters below the euphotic zone (0.00 ‰ at 150 m). The \$\delta^{56}\text{Fe}_{\text{diss}}\$ composition for
surface waters at this site are also isotopically distinct (\$p < 0.005\$ ) to \$\delta^{56}\text{Fe}_{\text{diss}}\$ values measured at the
SAZ and SOTS stations (Figure 3). At these stations, euphotic zone \$\delta^{56}\text{Fe}_{\text{diss}}\$ and \$\delta^{56}\text{Fe}_{\text{part}}\$ are
isotopically similar to each other, but significantly lower (\$p < 0.005\$ ) than in-eddy \$\delta^{56}\text{Fe}_{\text{diss}}\$ values
(Figure 3).

The heavy in-eddy $\delta^{56}\text{Fe}_{\text{diss}}$ values for the euphotic zone are consistent with biological fractionation
during dFe acquisition by phytoplankton. Modelling of the $\delta^{56}\text{Fe}_{\text{diss}}$ dataset using a closed system
model produced isotope fractionation factors (ϵ) of -2.3 ‰ for samples collected in the euphotic
zone (15 to 100 m; Figure 4). Interestingly, the in-eddy $\delta^{56}\text{Fe}_{\text{part}}$ values for the euphotic zone are not
consistent with an instantaneous or an integrated isotope fractionation process associated with a
closed system model. Generally, the expectation is that as dFe is consumed, the pFe pool should
become isotopically heavier for both the instantaneous and the accumulated product. While $\delta^{56}\text{Fe}_{\text{part}}$
does appear to be subtly heavier with decreasing dFe concentration, it is not consistent with closed-
system dependency for the biological reduction of Fe(III) to Fe(II) by cells (Figure 4). The physical
cycling of $\delta^{56}\text{Fe}_{\text{part}}$ and $\delta^{56}\text{Fe}_{\text{diss}}$ offers a possible explanation for this discrepancy: $\delta^{56}\text{Fe}_{\text{part}}$ is
distributed downward through the euphotic zone without general modification by sinking, whereas
changes in the $\delta^{56}\text{Fe}_{\text{diss}}$ with depth (or dFe concentration) across the euphotic likely represent mixing
across the euphotic zone.

Deleted: 9

Deleted: were

Deleted: ,

Using a generalised 1-D model, we simulated these two different mechanisms over the seasonal
cycle and found that the $\delta^{56}\text{Fe}_{\text{diss}}$ signal could be adequately modelled using isotope fractionation
associated with just dFe uptake alone ($\epsilon = -1 \text{ ‰}$) or a combination of isotope fractionation processes,
namely dFe uptake ($\epsilon = -0.6 \text{ ‰}$), pFe regeneration ($\epsilon = +0.15 \text{ ‰}$), dFe scavenging from solution ($\epsilon = -$
0.3 ‰) and dFe complexation to natural organic ligands ($\epsilon = +0.6 \text{ ‰}$) (Figure 4 and S9, see
supplementary material for extended discussion). The overall value for \$\epsilon\$ is considerably smaller than
the expected value (between -2 and -3 ‰^{ref 23}) for biological reduction of Fe(III) to Fe(II) by cells,
suggesting that dFe isotope fractionation is likely associated with the rapid trafficking of Fe between
the dissolved pool and cells (Figure 4). The recycling of Fe between pools may also amplify the
$\delta^{56}\text{Fe}_{\text{diss}}$ signal. The model also produces $\delta^{56}\text{Fe}_{\text{part}}$ values within the range measured in the euphotic
zone, supporting the idea that mixing and particle sinking are the primary mechanisms responsible
for distributing the $\delta^{56}\text{Fe}_{\text{diss}}$ and $\delta^{56}\text{Fe}_{\text{part}}$ signals across the euphotic zone.

[revised manuscript text omitted]

Deleted: Poitrasson and Freydier³⁴. This procedure involved loading samples onto ~200 μL columns filled with the precleaned anion exchange resin AG-MP1 (Bio-Rad). Salts and other elements not of interest were eluted by passing 3x 1 mL of 6 mol L⁻¹ hydrochloric acid. Fe

Deleted: /

Deleted: a

Deleted: volume

Deleted: a

Deleted: sample

Deleted: The overall Fe blank associated with the

Deleted: of

Deleted: amounted to <0.4 ng based on a

Deleted: L sample.

Deleted: ^{32,33} where:

$$\delta^{56}Fe = \left(\frac{{}^{56}Fe/{}^{54}Fe_{sample}}{{}^{56}Fe/{}^{54}Fe_{IRMM-014}} - 1 \right) \times 1000 \quad (1)$$

 The overall instrumental error for dFe and pFe samples ranged between $\pm 0.04 \text{ ‰}$ and $\pm 0.64 \text{ ‰}$
 (2σ). For low concentration samples, the instrumental error increased with decreasing Fe
 concentration and was associated with instrumental noise (Figure S6)⁴¹. Multiple large volume (3x
 2L) extractions and analysis of an in-house seawater standard had a reproducibility of $0.76 \pm 0.07 \text{ ‰}$
 (mean $\pm 2x$ standard deviation). Multiple analysis of a particulate sample had a reproducibility of
 $0.15 \pm 0.06 \text{ ‰}$ (mean $\pm 2x$ standard deviation). Analysis of geological samples NOD-A-1 and BCR-2
 produced values of $-0.43 \pm 0.03 \text{ ‰}$ and $0.04 \pm 0.07 \text{ ‰}$ (mean $\pm 2x$ standard deviation), respectively,
 which were consistent with literature values of -0.42 ± 0.07 ^{ref 42} for NOD-A-1 and 0.03 ± 0.06 ^{ref 42} for
 BCR-2

Deleted: ³⁵. Multiple

Deleted:) at an average dFe concentration of $0.78 \pm 0.08 \text{ nmol kg}^{-1}$

Deleted: ³⁶

Deleted: ³⁶ for BCR-2.

Dissolved iron concentration for each sample was calculated using sample weight and the amount
 double spike added to the sample. This calculation is based on isotope dilution using the known
 proportion of ⁵⁸Fe in the ⁵⁷Fe–⁵⁸Fe double spike^{38,43}. Note that the dFe concentrations presented
 here were not blank corrected, thus, they represent an upper concentration bound.

*Isotope modelling*

The closed system equation for the isotopic evolution of dFe as it is consumed can be described as
 follows:

$$\delta^{56}Fe_{dissolved} = \delta^{56}Fe_{dissolved.100m} + \varepsilon \ln(f) \quad (2)$$

 Where ε represents the isotope enrichment factor between the product (biologically utilised Fe) and
 the substrate (dFe) and f presents the fraction of dFe relative to the concentration of dFe at 100 m.
 The evolution of the instantaneous or integrated isotope fractionation processes can be modelled
 using the following expressions:

$$\delta^{56}Fe_{particulate} = \delta^{56}Fe_{dissolved} - \varepsilon \quad (3)$$

$$\delta^{56}Fe_{particulate} = \delta^{56}Fe_{dissolved.100m} + \frac{\varepsilon \ln(f)}{1-f} \quad (4)$$

*1D biogeochemical modelling*

The potential processes that influence the distribution and isotope fractionation of dFe and pFe
were explored using a 1D model (Figure S7). The rationale for using this 1D model is to explore the
relative influence (and interplay) of processes such as phytoplankton utilisation of Fe, its
complexation to natural organic ligands, its regeneration from sinking organic matter and the role of
scavenging on distribution and expression of isotope profiles. The model is based on Schlosser, et al.
⁴⁴ and includes one phytoplankton group and references key nutrients including nitrate, phosphate
and Fe (Figures S7, S8 and S9). The model includes mixing, which supplies nutrients into the euphotic
zone and the main loss process for nutrients and Fe from the euphotic zone (organic matter export).
The Fe component in the model also includes complexation to natural organic ligands, scavenging
and the atmospheric supply of Fe through the deposition and dissolution of dust (Support
Information, Figure S7)). The model also does not include advection, which we justify for several
reasons: i) vertical advection, i.e. upwelling, occurs in the Southern Ocean primarily south of the
Polar Front and not in the SAZ and SAF regions examined here for the cold core eddy⁴⁵, ii) latitudinal
advection supplies waters with similar properties from upstream in the Antarctic Circumpolar
Current⁴⁶, and can thus be ignored; and iii) transport is dominated by northward Ekman transport,
and while this does supply nutrients over the annual mean, in late summer surface concentrations
between the SAF and PFZ are very uniform⁴⁷, so this term can also be neglected. The equations and
values associated with each biogeochemical process are presented in the Supporting materials and
in Tables S1 and S2.

Word count for methods section 2267

**Acknowledgments**

This research was financially supported under Australian Research Council's Discovery program
(DP170102108; DP130100679) and ship time from Australia's Marine National Facility. We are
grateful to the officers, crew, and research staff of the Marine National Facility *R.V. Investigator* for
their help with sample collection and generation of hydrochemistry data. We are grateful to Stacy
Deppeler for running the flow cytometry samples. We are grateful for the comments from three
anonymous reviewers for their thoughtful comments that helped to improve the manuscript.

**Author contributions**

Deleted: Schlosser, et al. ³⁷

Deleted: ³⁸

Deleted: ³⁹

Deleted: ⁴⁰

Deleted: 1672

488 M.J.E. and P.W.B. conceived the study with input from P.G.S. and T.W.T. on appropriate
oceanographic sampling sites. M.J.E., R.F.S. and M.F. collected and analysed samples. M.J.E. wrote
the paper with assistance from all co-authors.

**Table**

**Table 1.** Intracellular iron:carbon (Fe:C) ratios, Fe uptake rates and dissolved Fe concentrations for upper Southern Ocean waters between longitudes 142°E
and 172°E.

Station/Study Latitude	Fe:C ($\mu\text{mol mol}^{-1}$)	Fe uptake rate ($\text{pmol L}^{-1} \text{d}^{-1}$)	Dissolved Fe (pmol kg^{-1})	Notes	Reference
Cold core eddy - 49.7°S	224 ± 38	25 ± 5	23 ± 1	15-40 m - 80-20% irradiance	this study
SAZ - 51.4°S	40 ± 8	12 ± 2	48 ± 17	15-40 m - 80-20% irradiance	this study
SOTS - 46.7°S	61 ± 7	28 ± 6	60 ± 4	15-40 m - 80-20% irradiance	this study
SOIREE - 61°S	3-7.5 [#]	3.07 [#]	80 ± 30 [§]	Fe enrichment experiment	Bowie et al. ⁴⁸
FeCycle - 46°S	5.5 - 19	290-360	51 ± 11	Subantarctic experiment	McKay et al. ⁴⁹
SOFex - 56°S	9-11	not reported	140	Northern Patch	Twining et al. ⁵⁰
SAZ Project - 47°S	52	60	70	SOTS station	Boyd et al. ¹⁸
SAZ Project - 54°S	78	55	70	Polar water station - equivalent to where the cyclonic eddy originated	Boyd et al. ¹⁸
SAZ-Sense P1 - 46.3°S	70 ± 44	110 ± 10	260 ± 40	Equivalent to SOTS station	Bowie et al. ⁵¹
SAZ-Sense P2 - 54°S	60 ± 9	34 ± 5	210 ± 20	Polar water station - equivalent to where the cyclonic eddy originated	Bowie et al. ⁵¹
SAZ-Sense P3 - 45.5°S	74 ± 47	77 ± 10	440 ± 70	Site close to Subtropical convergence	Bowie et al. ⁵¹

[§] Background dissolved Fe concentration before Fe infusion. [#] Fe:C and Fe uptake rates are for days 1-3 after Fe infusion.

Formatted: Font color: Auto

Deleted: Bowie et al. ⁴¹Deleted: McKay et al. ⁴²Deleted: Twining et al. ⁴³Deleted: Boyd et al. ¹²Deleted: Boyd et al. ¹²Deleted: Bowie et al. ⁴⁴Deleted: Bowie et al. ⁴⁴Deleted: Bowie et al. ⁴⁴

**Figure captions**

**Figure 1.** Chlorophyll and sea surface temperature in the study area. **a** and **b**, Maps of chlorophyll *a*
concentration **c**. Sea Surface Temperature (SST) and **d**. Sea Level Anomaly (SLA) for Southern Ocean
waters south of Australia. The diamonds represent the Cold Core eddy station (CCE), the
Subantarctic zone station (SAZ) and the Southern Ocean Time Series station (SOTS) and the solid
black line represents the Triaxus tow (Figure S1). The chlorophyll *a* satellite data represents a
monthly average for March 2016. The SST data are for 3 April 2016 and SLA data are for 25 March
2016. In **b** and **c** the contours lines are SST. Data extracted from
<https://coastwatch.pfeg.noaa.gov/erddap/griddap/>.

**Figure 2.** Depth profiles of Fe and carbon uptake. Profiles of **a**, Fe and **b**, carbon uptake versus depth.
**c**, Profiles of the intracellular Fe:C uptake ratio for the CCE, SAZ and SOTS stations.

**Figure 3.** Depth profiles of Fe concentration and isotope composition. Upper ocean (0 to 600 m)
depth profiles of dFe and pFe concentration (**a**, **b**, **c**), and the isotope composition (**d**, **e**, **f**) for
samples collected at the **a**, **d**., CCE, **b**., **e**., SAZ and **c**., **f**., SOTS stations.

**Figure 4.** Iron isotope fractionation model calculations. **a**. Dissolved $\delta^{56}\text{Fe}$ values for samples
collected at the CCE, SAZ and SOTS stations versus Fe concentration. **b**. Model curves for closed
steady-state isotope fractionation (equations 2 to 4) of dFe for the CCE. The best fit ϵ value for the
cold core eddy dFe data is -2.3 ‰. One-D model profiles (blue lines) for dFe and pFe versus depth for
**c**. ϵ equal to -1.0 ‰ ($\alpha_{\text{uptake}} = 0.999$) and **d**. ϵ equal to -2.3 ‰ ($\alpha_{\text{uptake}} = 0.9977$) along with profiles of
dFe and pFe isotope composition versus depth for the CCE.

Deleted: and

Deleted: surface temperature

Deleted: Diamonds

Deleted: represent

Deleted: . The white

**References**

[revised manuscript text omitted]

Deleted: 37

Deleted: 38

Deleted: 39

Deleted: 40

Deleted: 41

Deleted: 42

Deleted: 43

Deleted: 44

721

Reviewers' comments:

Reviewer #1 (Remarks to the Author):

Thank you for addressing my comments. From my (non-iron but eddy-focused) perspective, I am fine with the publication of the manuscript as it is.

Grammar:

* L37ff: The sentence "These eddies can have closed circulation thus 'trapping' the biogeochemical properties of these water such as nutrients, chlorophyll and particle concentrations differ compared to external waters. " needs correction.

Reviewer #2 (Remarks to the Author):

I can't shake the feeling that the Fe isotopes data could possibly be incorrect. Seawater Fe isotopes are quite challenging to measure, and the authors are presenting $\delta^{56}\text{Fe}$ for some of the lowest-Fe (and thus most challenging) waters ever reported. Also, they present a blank for purification of Fe on AG-MP1 resin which is lower than ever reported previously, by a factor of something like 5-10 by my rough estimation, without any obvious reason why their blank should be so much lower than other studies. Also, their reported blank of 0.4 ± 0.32 ng Fe suggests great variability in this value, and thus uncertainty in $\delta^{56}\text{Fe}$.

The key "gold standard" experiment which would allay my analytical concerns would be to remove Fe from seawater, then dope back in a small amount of non-crustal Fe standard (e.g. 50 pM Fe which is either +2 permil or - 2 permil), then extract that new standard Fe and show that the correct $\delta^{56}\text{Fe}$ value is obtained. Successfully performing such an experiment would be very convincing, and indeed such experiments are often presented by labs which are developing new seawater metal isotope methods. However, I recognize that, depending on whether they are still set up to measure Fe isotopes routinely, doing such experiments could require a fair amount of work, and it seems unfair to hold up publication of a paper until a whole new round of methods-development can be undertaken.

Alternatively, perhaps the authors could more honestly discuss the incredible challenges of measuring Fe at such low concentrations, and acknowledge that they have not yet done the "gold standard" experiment of doping an isotope standard back into Fe-free seawater (as described above). They could also provide even more information about issues such as the variability in their blank. Then they could discuss the reasons why they still believe that their data are sound, and reasons why any expected small errors in their measurement should not undermine their basic conclusions.

I appreciate that the authors have responded in a detailed fashion to my earlier concerns about methods. They have made a convincing argument that their Fe radioisotope uptake experiments and Fe concentration data are valid. And if the reported blanks and other methods details are taken at face value, then their $\delta^{56}\text{Fe}$ data should also be correct. The interpretation of the data is reasonable and the conclusions are significant.

I support eventual publication of this manuscript, and I don't wish to ask for an insurmountable amount of additional effort. If the authors are able to perform the "gold standard" experiment, that would greatly strengthen this manuscript, and it would clearly establish them as a lab capable of measuring Fe isotopes even in low-Fe Southern Ocean waters, which in turn would open up exciting new areas of research. Alternatively, I hope that an extraordinarily thorough and honest discussion of the strengths, and possible weaknesses, of their $\delta^{56}\text{Fe}$ methods can be included, as a step towards the eventual goal of establishing unequivocally how Fe isotopes cycle in the Southern Ocean.

Best,

Seth John

Reviewer #3 (Remarks to the Author):

I am happy the authors have addressed my comments.

Dear Dr Frischkorn,

We are grateful to all three reviewers for their comments on our manuscript. Below is our response to points raised by the reviewres.

Reviewers' comments:

Reviewer #1 (Remarks to the Author):

Thank you for addressing my comments. From my (non-iron but eddy-focused) perspective, I am fine with the publication of the manuscript as it is.

Grammar:

* L37ff: The sentence "These eddies can have closed circulation thus 'trapping' the biogeochemical properties of these water such as nutrients, chlorophyll and particle concentrations differ compared to external waters. " needs correction.

Response: We have revised this sentence so it now reads "These eddies can have closed circulation thus 'trapping' ref 9 the biogeochemical properties of these features such that nutrient, chlorophyll and particle concentrations can be distinct relative to those in the surrounding waters^{2,9-11}"

Reviewer #2 (Remarks to the Author):

I can't shake the feeling that the Fe isotopes data could possibly be incorrect. Seawater Fe isotopes are quite challenging to measure, and the authors are presenting $d_{56}\text{Fe}$ for some of the lowest-Fe (and thus most challenging) waters ever reported. Also, they present a blank for purification of Fe on AG-MP1 resin which is lower than ever reported previously, by a factor of something like 5-10 by my rough estimation, without any obvious reason why their blank should be so much lower than other studies. Also, their reported blank of 0.4 ± 0.32 ng Fe suggests great variability in this value, and thus uncertainty in $d_{56}\text{Fe}$.

Response: In 2019 I (Michael Ellwood) published a paper describing iron isotope transformations in Lake Cadagno Switzerland (Ellwood et al., 2019). In that study, the anion exchange resin AG-MP1 was also utilised to separate iron from other ions for a freshwater study for a Swiss mountain lake. This work was undertaken in the labs at ETH Zurich and utilised their chemicals and their resin. The overall dissolved iron procedural blank of that study was 0.6 ± 0.5 ng (Ellwood et al., 2019), which included evaporating samples to dryness and passing the samples over the AG-MP1 resin. The blank from that study is comparable to the blanks results obtained in the present study (0.40 ± 0.32 ng) where column separation of iron was undertaken at the ANU using a different batch on AG-MP1 resin to that used at ETH. Subsequent to the analysis of the 2016 samples (i.e., results featured in our manuscript), we have managed to obtain the same, and in some instances lower, blank values for the AG-MP1 resin for processing other iron isotope samples. Thus, in the present study we

present a blank that is comparable to that reported previously, and which has subsequently been repeatable (and surpassed) at the ANU.

The key “gold standard” experiment which would allay my analytical concerns would be to remove Fe from seawater, then dope back in a small amount of non-crustal Fe standard (e.g. 50 pM Fe which is either +2 permil or - 2 permil), then extract that new standard Fe and show that the correct $\delta^{56}\text{Fe}$ value is obtained. Successfully performing such an experiment would be very convincing, and indeed such experiments are often presented by labs which are developing new seawater metal isotope methods. However, I recognize that, depending on whether they are still set up to measure Fe isotopes routinely, doing such experiments could require a fair amount of work, and it seems unfair to hold up publication of a paper until a whole new round of methods-development can be undertaken.

Response: Prior to the analysis of the samples from the 2016 voyage, we did indeed test that the ANU method was producing high-quality results. This was done by undertaking an inter-calibration exercise (perhaps the “platinum standard” as each analysis is completely independent) with Tim Conway and Matthies Sieber at ETH. In this inter-calibration exercise, samples from the GEOTRACES crossover station at 32.5°S, 170°W for the GP13 were analysed by the ETH group. The ANU results were also compared to iron isotope results generated by the ETH for samples collected GP19 voyage at 32.5°S, 170°W crossover station. The results from the two groups showed that there were no systematic differences between methods used by the two groups – see the figure below. The results from this inter-calibration exercise were also presented at the 2018 Ocean Sciences meeting (Conway et al., 2018), and Conway et al., are in the process of preparing a manuscript detailing the results from this inter-calibration exercise.

Prior to developing the double spike method at the ANU, the ANU group also participated in the GEOTRACES inter-calibration exercise where groups were able to analyse samples collected from the BATS site near Bermuda. The results produced by the ANU group using the standard-sample-standard bracketing technique were comparable to the other groups undertaking iron isotope analysis (see the table below). This work was published by Boyle et al. (2012).

Overall, the ANU double spike method produces iron isotope results that are comparable to other field-leading international groups working in the space.

Table 2. Fe isotope data for GEOTRACES IC1 samples.

Sample	Depth (m)	ANU $\delta^{56}\text{Fe}/^{54}\text{Fe}$	WHOI $\delta^{56}\text{Fe}/^{54}\text{Fe}$	Caltech $\delta^{56}\text{Fe}/^{54}\text{Fe}$	LEGOS $\delta^{56}\text{Fe}/^{54}\text{Fe}$	LEGOS [Fe] (nmol/kg)
GSI	7	0.32±0.06 (2se)	+0.24±0.10 (2se n=3)	+0.32±0.06 (2se n=2)	+0.41±0.04 (2se n=3)	0.42
GDI	2000	0.45±0.13 (2se)	+0.42±0.11 (2se n=3)	+0.55±0.03 (2se n=5)	+0.52±0.07 (2se n=2)	0.84
GPrI	75			+0.41±0.06 (2se n=4)		
GPrI	125			+0.30±0.06 (2se n=5)		
GPrI	250			+0.45±0.05 (2se n=3)		
GPrI	500			+0.34±0.05 (2se n=2)		
GPrI	1000			+0.35±0.05 (2se n=2)		
GPrI	1500			+0.35±0.05 (2se n=2)		
GPrI	2500			+0.71±0.05 (2se n=2)		
4200m (individual sample)	4200			+0.35±0.07 (2se n=1)		

All $\delta^{56}\text{Fe}/^{54}\text{Fe}$ data are per mill relative to IRMM-014

Table 2 taken from Boyle et al. (2012) with the ANU results highlighted.

The above Figure presents the iron isotope results for samples collected on the GP13 and GP19 campaigns for a crossover station located at 32.5°S, 170°W in the South Pacific Ocean. The iron concentration results (left panel) are from the ANU group and have been published (Ellwood et al., 2018). The iron isotope results are for samples collected and analysed by the ANU group, samples the ANU group shared with the ETH group and samples collected on GP19 and analysed by the ETH group. One sample is highlighted as an outlier. The trends seen in the GP13 and GP19 profiles are consistent with each other and there is no systematic offset in the datasets, highlighting the quality of both the ANU and ETH methods.

Alternatively, perhaps the authors could more honestly discuss the incredible challenges of measuring Fe at such low concentrations, and acknowledge that they have not yet done the “gold standard” experiment of doping an isotope standard back into Fe-free seawater (as described above). They could also provide even more information about issues such as the variability in their blank. Then they could discuss the reasons why they still believe that their data are sound, and reasons why any expected small errors in their measurement should not undermine their basic conclusions.

Response: We have included the following sentences in the Methods section of the manuscript to highlight the quality of the iron isotope method. “The performance of the iron isotope method was also assessed through an inter-calibration exercise for samples from the GEOTRACES GP13 and GP19 campaigns at a crossover station located at 30°S; 170°W. The iron isotope results from the exercise were comparable – i.e., the trends seen in the GP13 and GP19 profiles are consistent with each other⁴³.”

I appreciate that the authors have responded in a detailed fashion to my earlier concerns about methods. They have made a convincing argument that their Fe radioisotope uptake experiments and Fe concentration data are valid. And if the reported blanks and other methods details are taken at face value, then their $\delta^{56}\text{Fe}$ data should also be correct. The interpretation of the data is reasonable and the conclusions are significant.

Response: Thank you

I support eventual publication of this manuscript, and I don't wish to ask for an insurmountable amount of additional effort. If the authors are able to perform the "gold standard" experiment, that would greatly strengthen this manuscript, and it would clearly establish them as a lab capable of measuring Fe isotopes even in low-Fe Southern Ocean waters, which in turn would open up exciting new areas of research. Alternatively, I hope that an extraordinarily thorough and honest discussion of the strengths, and possible weaknesses, of their $\delta^{56}\text{Fe}$ methods can be included, as a step towards the eventual goal of establishing unequivocally how Fe isotopes cycle in the Southern Ocean.

We have added the following sentence to the manuscript and point the reader to additional figure (S10) in the support materials highlighting the quality of the data. "The performance of the Fe isotope method was also assessed through an intercalibration exercise for samples from the GP13 and GP19 campaigns at a crossover station located at 30°S; 170°W. The iron isotope results from the exercise were comparable – i.e., the trends seen in the GP13 and GP19 profiles are consistent with each other (Figure S10) ⁴³"

Best,

Seth John

Reviewer #3 (Remarks to the Author):

I am happy the authors have addressed my comments.

Thank you

References

- Boyle, E.A., John, S., Abouchami, W., Adkins, J.F., Echegoyen-Sanz, Y., Ellwood, M., Flegal, A.R., Fornace, K., Gallon, C., Galer, S., Gault-Ringold, M., Lacan, F., Radic, A., Rehkamper, M., Rouxel, O., Sohrin, Y., Stirling, C., Thompson, C., Vance, D., Xue, Z. and Zhao, Y., 2012. GEOTRACES IC1 (BATS) contamination-prone trace element isotopes Cd, Fe, Pb, Zn, Cu, and Mo intercalibration. *Limnology and Oceanography: Methods*, 10: 653-665.
- Conway, T.M., Sieber, M., Ellwood, M.J., Takano, S., Sohrin, Y. and Vance, D., 2018. CT31A-03: The competing influence of local cycling, regional sources and Southern Ocean processes in

influencing Fe isotope cycling at lower latitudes in the oceans, *Ocean Sciences*, Portland, Oregon.

Ellwood, M.J., Bowie, A.R., Baker, A., Gault-Ringold, M., Hassler, C., Law, C.S., Maher, W.A., Marriner, A., Nodder, S., Sander, S., Stevens, C., Townsend, A., van der Merwe, P., Woodward, E.M.S., Wuttig, K. and Boyd, P.W., 2018. Insights Into the Biogeochemical Cycling of Iron, Nitrate, and Phosphate Across a 5,300 km South Pacific Zonal Section (153°E–150°W). *Global Biogeochemical Cycles*: 2017GB005736.

Ellwood, M.J., Hassler, C., Moisset, S., Pascal, L., Danza, F., Peduzzi, S., Tonolla, M. and Vance, D., 2019. Iron isotope transformations in the meromictic Lake Cadagno. *Geochimica et Cosmochimica Acta*, 255: 205-221.

Reviewers' comments:

Reviewer #2 (Remarks to the Author):

For my last review, I was encouraged by the editor to present a complete and honest description of my concerns about the potential for inaccurate data when pushing Fe isotopes methods to report data on lower-Fe samples than previously reported.

I suggested that if the authors didn't want to perform more methods development, then they might:

"honestly discuss the incredible challenges of measuring Fe at such low concentrations, and acknowledge that they have not yet done the "gold standard" experiment of doping an isotope standard back into Fe-free seawater (as described above). They could also provide even more information about issues such as the variability in their blank. Then they could discuss the reasons why they still believe that their data are sound, and reasons why any expected small errors in their measurement should not undermine their basic conclusions."

In response they added two sentences stating that their methods perform well in intercalibration. In not adding more detail, I think the authors are missing an opportunity to move the field forwards in terms of understanding how far we can push Fe isotope measurements for low-Fe waters, which is extraordinarily important considering that such HNLC waters are exactly where Fe is a limiting nutrient. I don't mean to imply that the measurements presented here are somehow lagging behind the data quality of other labs (indeed their responses demonstrate that their data matches that of other top labs, see below), but just to point out that they are reporting $\delta^{56}\text{Fe}$ for waters with roughly an order of magnitude less Fe than anything else previously reported. In my opinion, discussing the challenges and successes of such measurements is an important and useful part of such a boundary-pushing effort.

While the authors have provided a great deal of additional information, which I appreciate, it doesn't quite get to the heart of my concern. The authors have demonstrated that they can perform Fe isotope analyses with skill just as high as other international labs. And they have demonstrated through intercalibration that they are able to accurately measure $\delta^{56}\text{Fe}$ on relatively high-concentration ($> \sim 0.2$ nM) samples. But I'm still not totally convinced that they (or anybody!) can accurately measure $\delta^{56}\text{Fe}$ on samples with tens-of-picomolar concentrations of Fe. All of the intercomparison exercises and plots they present deal with samples containing several hundred picomolar Fe. I would argue that the "platinum standard" intercomparison exercises are appropriate for showing that their methods are useful for hundreds-of-picomolar Fe, but that the "gold standard" re-doping experiments are really the only way to show whether or not it is possible to accurately measure $\delta^{56}\text{Fe}$ on tens-of-picomolar Fe. Of course, realizing the difficulty of such experiments, I also suggested that they might provide more detail in the manuscript about their method's possible strengths and weaknesses. With a little more detail, I think this manuscript will not only contribute the main points of their paper, but also contribute to an understanding of these methods.

All that said, I do realize that we are now discussing tiny details about the discussion of methods in this manuscript. This is not central to the overall message and value of the work, which I appreciate, and I do honestly hope it can be published soon.

Seth

Dear Dr Frischkorn,

Below in blue text is our rejoinder to comments raised by reviewer 2.

Regards

Michael

Reviewer #2 (Remarks to the Author):

For my last review, I was encouraged by the editor to present a complete and honest description of my concerns about the potential for inaccurate data when pushing Fe isotopes methods to report data on lower-Fe samples than previously reported.

I suggested that if the authors didn't want to perform more methods development, then they might:

"honestly discuss the incredible challenges of measuring Fe at such low concentrations, and acknowledge that they have not yet done the "gold standard" experiment of doping an isotope standard back into Fe-free seawater (as described above). They could also provide even more information about issues such as the variability in their blank. Then they could discuss the reasons why they still believe that their data are sound, and reasons why any expected small errors in their measurement should not undermine their basic conclusions."

In response they added two sentences stating that their methods perform well in intercalibration. In not adding more detail, I think the authors are missing an opportunity to move the field forwards in terms of understanding how far we can push Fe isotope measurements for low-Fe waters, which is extraordinarily important considering that such HNLC waters are exactly where Fe is a limiting nutrient. I don't mean to imply that the measurements presented here are somehow lagging behind the data quality of other labs (indeed their responses demonstrate that their data matches that of other top labs, see below), but just to point out that they are reporting $d^{56}\text{Fe}$ for waters with roughly an order of magnitude less Fe than anything else previously reported. In my opinion, discussing the challenges and successes of such measurements is an important and useful part of such a boundary-pushing effort.

In response, we have added a paragraph to the manuscript highlighting the challenges to making iron isotope measurements on low concentration samples. see lines 335 to 365.

"As with all open ocean seawater work, during the collection and processing of samples contamination can hinder the production of accurate and meaningful data. The added challenge for Fe isotope studies, particularly for low concentration systems such as the Southern Ocean, is obtaining enough material for isotope analysis. For the result presented here, the dFe processing blank associated represents as much $20 \pm 17\%$ of the concentration and the isotope signal. While concentration uncertainties are highest for shallow samples collected in the CCE, the structure of the dFe concentration versus depth profile for this station, and indeed the other two stations, are oceanographically consistent, i.e. they have low surface water concentrations that increase with depth⁴⁵. In a companion study, dissolved zinc concentration and zinc isotope results obtained from the same samples showed no indication of trace metal contamination associated with sample collection and processing⁴⁶. For the dFe isotope results, there is also the added challenge of obtaining enough material for isotope analysis. Here we optimised the isotopic measurement of dFe by reducing the volume of each sample presented for analysis (0.3 to 0.35 mL) thereby upping its

concentration to reduce errors associated with instrument noise^{41,47}. We also utilised a sample-spike ratio of ~1.6 (spike ⁵⁷Fe-⁵⁸Fe ratio = 1.05) such that counting errors are minimised for ⁵⁶Fe, ⁵⁷Fe, and ⁵⁸Fe. Even with these steps, the influence of instrument noise increased for low concentration Fe samples (Figure S6). While the uncertainty window around these measurements is larger than that for samples with a higher dFe concentration, the upper water column variations for $\delta^{56}\text{Fe}_{\text{diss}}$ between 15 and 150 m are statistically distinct and oceanographically consistent. The enrichment of $\delta^{56}\text{Fe}_{\text{diss}}$ within the euphotic zone is consistent with measurements made at 32.5°S, 150°W (Figure S10) and other recent measurements for low dFe concentration waters of the Southern Ocean⁴⁸. Likewise, the decline in $\delta^{56}\text{Fe}_{\text{diss}}$ values below the euphotic zone is consistent with measurements made at 32.5°S, 150°W (Figure S10), although one should be mindful that this station is outside of the Southern Ocean such that the biological community leading to variation in $\delta^{56}\text{Fe}_{\text{diss}}$ is likely to be different.”

While the authors have provided a great deal of additional information, which I appreciate, it doesn't quite get to the heart of my concern. The authors have demonstrated that they can perform Fe isotope analyses with skill just as high as other international labs. And they have demonstrated through intercalibration that they are able to accurately measure $\delta^{56}\text{Fe}$ on relatively high-concentration (>~0.2 nM) samples. But I'm still not totally convinced that they (or anybody!) can accurately measure $\delta^{56}\text{Fe}$ on samples with tens-of-picomolar concentrations of Fe. All of the intercomparison exercises and plots they present deal with samples containing several hundred picomolar Fe. I would argue that the "platinum standard" intercomparison exercises are appropriate for showing that their methods are useful for hundreds-of-picomolar Fe, but that the "gold standard" re-doping experiments are really the only way to show whether or not it is possible to accurately measure $\delta^{56}\text{Fe}$ on tens-of-picomolar Fe. Of course, realizing the difficulty of such experiments, I also suggested that they might provide more detail in the manuscript about their method's possible strengths and weaknesses. With a little more detail, I think this manuscript will not only contribute the main points of their paper, but also contribute to an understanding of these methods.

We would like to point out that the intercalibration study actually covered the following concentration range 0.017 and 0.72 nmol kg⁻¹. Indeed for samples shallower than 150 m were below 100 pmol kg⁻¹ with the lowest being 17 pmol kg⁻¹. We have pointed this out too. That said, in the added paragraph (line 335 onwards), we highlight the challenges with the method in overcoming instrument noise which increases isotopic errors. We have also justified why we think our dissolve iron isotope dataset is reasonable.

All that said, I do realize that we are now discussing tiny details about the discussion of methods in this manuscript. This is not central to the overall message and value of the work, which I appreciate, and I do honestly hope it can be published soon.

Seth

**.Distinct iron cycling in a Southern Ocean eddy**

Michael J. Ellwood*¹, Robert F. Strzepek², Peter G. Strutton^{2,3}, Thomas W. Trull⁴, Marion Fourquez²
and Philip W. Boyd².

1. Research School of Earth Sciences, Australian National University, Canberra, Australia

2. Institute for Marine and Antarctic Studies, University of Tasmania, Hobart, Australia

3. Australian Research Council Centre of Excellence for Climate Extremes, University of Tasmania,
Hobart, Australia

4. CSIRO Oceans and Atmosphere, Hobart, Australia

10Dec19 version

Deleted: 20Nov19

**Abstract**

Mesoscale eddies are ubiquitous in the ocean, controlling ocean-atmosphere heat and gas
exchange. However, their physical influence on phytoplankton production in the iron-limited
Southern Ocean remains unknown. Here, we probed iron (Fe) cycling in a subantarctic cold-core
eddy during the austral autumn. In-eddy, surface dissolved Fe concentrations and phytoplankton
productivity and biomass were exceedingly low relative of external waters. In contrast, in-eddy
phytoplankton Fe-to-carbon uptake ratios were elevated 2-6 fold, indicating upregulated
intracellular Fe acquisition, thus reducing the residence time of Fe in the dissolved pool to ~1
24 day. Dissolved Fe isotope values were isotopically heavy in the euphotic zone, indicative of
25 extensive trafficking of Fe by cells within the eddy. Below the euphotic zone, Fe isotope values
were lighter and coincident with peaks in recycled nutrients and cell abundance, again indicative
of enhanced microbially-mediated Fe recycling. Our measurements show that the isolated
nature of eddies can produce distinctly different Fe biogeochemistry compared to surrounding
waters with cells upregulating iron uptake and using recycling processes to sustain themselves.
Recognising these eddy properties is essential to understanding how they contribute to the
biophysicochemical structure of the Southern Ocean.

Words = 187

**Main text**

[revised manuscript text omitted]
 (Figure 3). In the euphotic zone (0-85m) of the cold core eddy, $\delta^{56}\text{Fe}_{\text{diss}}$ values are
129 isotopically heavy at $+1.19 \text{ } \text{‰}$ and distinct ($p < 0.005$, T-test) to that of pFe ($\delta^{56}\text{Fe}_{\text{part}}$) and the $\delta^{56}\text{Fe}_{\text{diss}}$
composition for waters below the euphotic zone ($0.00 \text{ } \text{‰}$ at 150 m). The $\delta^{56}\text{Fe}_{\text{diss}}$ composition for
surface waters at this site are also isotopically distinct ($p < 0.005$) to $\delta^{56}\text{Fe}_{\text{diss}}$ values measured at the
SAZ and SOTS stations (Figure 3). At these stations, euphotic zone $\delta^{56}\text{Fe}_{\text{diss}}$ and $\delta^{56}\text{Fe}_{\text{part}}$ are
isotopically similar to each other but significantly lower ($p < 0.005$) than in-eddy $\delta^{56}\text{Fe}_{\text{diss}}$ values
(Figure 3).

The heavy in-eddy $\delta^{56}\text{Fe}_{\text{diss}}$ values for the euphotic zone are consistent with biological fractionation
during dFe acquisition by phytoplankton. Modelling of the $\delta^{56}\text{Fe}_{\text{diss}}$ dataset using a closed system
model produced isotope fractionation factors (ϵ) of $-2.3 \text{ } \text{‰}$ for samples collected in the euphotic
zone (15 to 100 m; Figure 4). Interestingly, the in-eddy $\delta^{56}\text{Fe}_{\text{
[revised manuscript text omitted]

Deleted: campaigns

Formatted: Font color: Text 1

Formatted: Font color: Text 1

Deleted: .

Formatted: Superscript

Formatted: Space After: 10 pt

Deleted:

Deleted: s

Deleted: for

Deleted: {Bruland, 1980 #3100}

concentration to reduce errors associated with instrument noise^{41,47}. We also utilised a sample-spike
 ratio of ~1.6 (spike ⁵⁷Fe-⁵⁸Fe ratio = 1.05) such that counting errors are minimised for ⁵⁶Fe, ⁵⁷Fe, and
 ⁵⁸Fe. Even with these steps, the influence of instrument noise increased for low concentration Fe
 samples (Figure S6). While the uncertainty window around these measurements is larger than that
 for samples with a higher dFe concentration, the upper water column variations for $\delta^{56}\text{Fe}_{\text{diss}}$
 between 15 and 150 m are statistically distinct and oceanographically consistent. The enrichment of
 $\delta^{56}\text{Fe}_{\text{diss}}$ within the euphotic zone is consistent with measurements made at 32.5°S, 150°W (Figure
 S10) and other recent measurements for low dFe concentration waters of the Southern Ocean⁴⁸.
 Likewise, the decline in $\delta^{56}\text{Fe}_{\text{diss}}$ values below the euphotic zone is consistent with measurements
 made at 32.5°S, 150°W (Figure S10), although one should be mindful that this station is outside of
 the Southern Ocean such that the biological community leading to variation in $\delta^{56}\text{Fe}_{\text{diss}}$ is likely to be
 different.

- Deleted: ⁴¹#5599⁴⁷
- Formatted: Superscript
- Formatted: Superscript
- Formatted: Superscript
- Formatted: Superscript
- Formatted: Superscript
- Deleted: levels
- Deleted: in

- Deleted: influencing
- Deleted:

*Iron isotope modelling*

The closed system equation for the isotopic evolution of dFe as it is consumed can be described as
 follows:

$$\delta^{56}\text{Fe}_{\text{dissolved}} = \delta^{56}\text{Fe}_{\text{dissolved.100m}} + \epsilon \ln(f) \quad (2)$$

 Where ϵ represents the isotope enrichment factor between the product (biologically utilised Fe) and
 the substrate (dFe) and f presents the fraction of dFe relative to the concentration of dFe at 100 m.
 The evolution of the instantaneous or integrated isotope fractionation processes can be modelled
 using the following expressions:

$$\delta^{56}\text{Fe}_{\text{particulate}} = \delta^{56}\text{Fe}_{\text{dissolved}} - \epsilon \quad (3)$$

$$\delta^{56}\text{Fe}_{\text{particulate}} = \delta^{56}\text{Fe}_{\text{dissolved}} + \frac{\epsilon \ln(f)}{1-f} \quad (4)$$

*1D biogeochemical modelling*

[revised manuscript text omitted]

**Figure 3.** Depth profiles of Fe concentration and isotope composition. Upper ocean (0 to 600 m)
depth profiles of dFe and pFe concentration (**a**, **b**, **c**), and the isotope composition (**d**, **e**, **f**) for
samples collected at the **a**, **d**., CCE, **b**., **e**., SAZ and **c**., **f**., SOTS stations. See the Methods section for
details on the issues and challenges involved with measuring \$\delta^{56}\text{Fe}\$ on samples with very low dFe
concentrations.

**Figure 4.** Iron isotope fractionation model calculations. **a**. Dissolved $\delta^{56}\text{
[revised manuscript text omitted]

*Global Biogeochem. Cycles* **23**, (2009).

594

REVIEWERS' COMMENTS:

Reviewer #2 (Remarks to the Author):

I thank the authors for their diligence in addressing my concerns, and I think this manuscript is suitable for publication.

Dear Dr Frischkorn,
Below in blue text is our rejoinder to comments raised by reviewer 2.
Regards
Michael

Reviewer #2 (Remarks to the Author):

I thank the authors for their diligence in addressing my concerns, and I think this manuscript is suitable for publication.

Thank you, your time and efforts in reviewing our manuscript is appreciated!